# Protective Immunity against *Listeria monocytogenes* in Rats, Provided by HCl- and NaOH-Induced *Listeria monocytogenes* Bacterial Ghosts (LMGs) as Vaccine Candidates

**DOI:** 10.3390/ijms23041946

**Published:** 2022-02-09

**Authors:** Seongmi Ji, Eun Sun Moon, Han Byul Noh, Hyun Jung Park, Seongdae Kim, Sung Oh, Nagarajan Vinod, Chang Won Choi, Kilhan Kwak

**Affiliations:** 1Department of Biology, Graduate School, Pai Chai University, Daejeon 35345, Korea; hijsm91@gmail.com (S.J.); ansdmstjs124@naver.com (E.S.M.); creator1018@pcu.ac.kr (H.B.N.); parkhj0524@bioneer.co.kr (H.J.P.); khboy111@pcu.ac.kr (S.K.); ohsung85@gmail.com (S.O.); biovinz@gmail.com (N.V.); 2Animal Health Institute of Jeollabukdo Western Branch, Jeonbuk Provincial Government, 18-4, Jangsu-gun 55632, Korea; kkilhan@korea.kr

**Keywords:** *Listeria monocytogenes*, bacterial ghosts, HCl-induced *Listeria monocytogenes* ghosts, NaOH-induced *Listeria monocytogenes* ghosts, cytotoxicity, pro- and anti-inflammatory mediators, vaccine candidate, humoral immunity, cell-mediated immunity, protection against virulent challenge

## Abstract

*Listeria monocytogenes* (*Lm*) bacterial ghosts (LMGs) were produced by the minimum inhibitory concentration (MIC) of HCl, H_2_SO_4_, and NaOH. Acid and alkali effects on the LMGs were compared by in vitro and in vivo analyses. Scanning electron microscope showed that all chemicals form lysis pores on the *Lm* cell envelopes. Real-time qPCR revealed a complete absence of genomic DNA in HCl- and H_2_SO_4_-induced LMGs but not in NaOH-induced LMGs. HCl-, H_2_SO_4_- and NaOH-induced LMGs showed weaker or missing protein bands on SDS-PAGE gel when compared to wild-type *Lm*. Murine macrophages exposed to the HCl-induced LMGs showed higher cell viability than those exposed to NaOH-induced LMGs or wild-type *Lm*. The maximum level of cytokine expression (TNF-α, iNOS, IFN-γ, and IL-10 mRNA) was observed in the macrophages exposed to NaOH-induced LMGs, while that of IL-1β mRNA was observed in the macrophages exposed to HCl-induced LMGs. To investigate LMGs as a vaccine candidate, mice were divided into PBS buffer-injected, HCl- and NaOH-induced LMGs immunized groups. Mice vaccinated with HCl- and NOH-induced LMGs, respectively, significantly increased in specific IgG antibodies, bactericidal activities of serum, and CD4^+^ and CD8^+^ T-cell population. Antigenic *Lm* proteins reacted with antisera against HCl- and NOH-induced LMGs, respectively. Bacterial loads in HCl- and NaOH-induced LMGs immunized mice were significantly lower than PBS-injected mice after virulent *Lm* challenges. It suggested that vaccination with LMGs induces both humoral and cell-mediated immune responses and protects against virulent challenges.

## 1. Introduction

Listeriosis, caused by the Gram-positive, facultative intracellular bacterium *Listeria monocytogenes* (*Lm*), is one of the major foodborne pathogens [1] with a high mortality rate of 20~30% [2,3]. *Lm* has an arsenal of virulence proteins in the cell envelope, and these factors are presumed to be specific receptors that facilitate its pathogenesis and cell infection cycle, making it easy to enter target cells, circumvent autophagy, and spread cell to cell [4,5,6]. Besides, *Lm* could invade and propagate in phagocytic and non-phagocytic cells through the forming internalization vesicle. Especially, in non-phagocytic cells, *Lm* can escape before lysosomal fusion and grow within the cytoplasm [7,8,9].

*Lm* adhesion to host cells could be mediated by *Listeria* adhesion protein (LAP) interacting with the host cell receptor heat shock protein 60 (Hsp60) [10]. Four major virulence factors are involved in host infection, i.e., actin polymerization protein (ActA) [11,12], internalin A (InlA) [12,13], internalin B (InlB) [12,13], and listeriolysin O (LLO) [14,15]. ActA mediates actin-based movement and cell-to-cell spread, enabling *Lm* to escape autophagy [11,12]. The LPXTG proteins were predicted to attach covalently to the peptidoglycan [16]. One of the typical LPXTG proteins is InlA that promotes bacterial entry into epithelial cells [12,13]. InlB having three C-terminal glycine and tryptophan (GW) repeat domains is also required for *Lm* entry into many eukaryotic cell types [12,13,16]. A synergistic two-step mechanism has been proposed: InlA provides the specificity of *Lm* adhesion to multicellular junctions at the villus tips, and InlB locally activates c-Met to accelerate junctional endocytosis and bacterial invasion of the intestine [17]. The pore-forming toxin, LLO, is a crucial virulence factor that facilitates the *Lm* escape from the phagosomal compartments into the cytoplasm of infected cells [14,15].

Recently, hypervirulent strains and multi-drugresistant isolates of *Lm* have emerged as a considerable threat to public health [18,19,20]. To date, no prophylactic vaccine against *Lm* is available, although vaccines are very effective in protecting hosts from bacterial infection [21]. Recently, a triple-virulence-gene deletion mutant of the vaccine strain stimulated higher anti-LLO antibodies, induced an effective Th1 type immune response, and provided 100% protection against serovar 4b and 1/2b challenge [21]. The MHC class I and II molecules efficiently delivered *Lm’s* antigenic peptides that bound to either CD4^+^-T or CD8^+^-T cell receptors [21,22,23,24]. In this respect, *Lm* has become a promising platform for a vaccine vehicle after the virulence attenuation. For cancer immunotherapy, researchers have developed novel recombinant *Lm* strains capable of expressing and secreting a variety of tumor-associated antigens, some of which have entered preclinical or clinical phases against a wide range of cancers [25,26,27].

Bacterial ghosts (BGs) have been proposed as a multivalent vaccine candidate conventionally produced by temperature-controlled expression of the cloned lysis gene *E* from bacteriophage PhiX174 [28,29]. BGs are non-living structures, possessing trans-membrane lysis tunnel structures on their cell envelope [30,31,32]. The resulting BGs retained basic bacterial cell surface structures but lacked cytoplasmic contents [33,34,35]. Such BGs have intrinsic adjuvant properties to induce humoral and cell-mediated immune responses against virulent challenges in various animal models [33,36]. Although *E* gene-induced BGs provide an efficient protective immunity against specific infections [37,38,39], the method is limited to Gram-negative bacteria and needs a multi-step process that is expensive and time-consuming. Besides, it is difficult to reach the 100% lysis rate (no viable cells) of target bacteria in a short time [40], which may cause potential risks.

Alternatively, sodium hydroxide (NaOH)-induced BGs were generated by using the minimum inhibitory concentration (MIC) from Gram-negative bacteria such as *Escherichia coli* [41], *Salmonella enterica* Enteritidis [42,43], *S. enterica* Typhimurium [44,45], *Vibrio parahaemolyticus* [46], and Gram-positive bacteria such as *Staphylococcus aureus* [47] and *Lm* [48]. Immunization with BGs induced effective immune responses and protected against virulent challenges [41,44,46]. Some acids have strong bactericidal effects, and their MICs were applied to produce BGs from a Gram-negative bacterium [42]. However, a detectable amount of DNA contents was amplified from the acid-induced BGs but not alkali-induced BGs [46]. The question arose regarding whether BGs from a Gram-positive bacterium would show similar antigenic properties to an acid treatment compared to alkali treatment.

In the present study, we initially compared HCl-, H_2_SO_4_-, and NaOH-induced *Listeria* BGs (LMGs) by morphological analysis, protein and DNA profiles, cytotoxicity, and quantitative analysis of cytokine mRNAs. To determine the efficiency of HCl- and NaOH-induced LMGs as a potential vaccine candidate, we compared abilities to induce protective immune responses against a virulent *Lm* challenge in BALB/c mice. Additionally, this study demonstrated that HCl- and NaOH-induced LMGs have several antigenic surface proteins located at the cell envelope.

## 2. Results and Discussion

### 2.1. Effects of Chemicals on Bacterial Cell Envelopes

The *Lm* culture (1 × 10^6^ CFU/mL) treated with HCl, H_2_SO_4_, and NaOH showed the MIC of 6.25, 12.5, and 6.25 mg/mL and the minimum bactericidal concentrations (MBC) of 12.5, 25.0, and 12.5 mg/mL, respectively (Table 1). The MIC of each chemical was used for LMG production.

### 2.2. Morphological Observation of Chemical-Induced LMGs by Scanning Electron Microscopy (SEM)

Compared with wild-type *Lm* cells, HCl-, H_2_SO_4_-, and NaOH-induced LMGs formed the trans-membrane lysis pores on the cell surfaces (Figure 1). Among chemical treatments, H_2_SO_4_ caused more severe damage to the cell walls and produced larger lysis pores than the HCl and NaOH. Depending upon the bacterial species, a wide range of chemicals could damage cellular envelopes, leading to the formation of pores and loss of intracellular components such as nucleic acids [49]. Like the lysis E gene-induced BGs, NaOH-induced BGs showed clear trans-membrane pore structures on the bacterial cell surfaces [42,43,45,46,47].

### 2.3. Analysis of DNA and Protein Profiles in Chemically Induced LMGs

DNA bands were not detected in HCl-, H_2_SO_4_- and NaOH-induced LMGs (Figure 2a–c, lane 2) compared with wild-type *Lm* cells (lane 1) on agarose gel electrophoresis. Besides, agarose gel electrophoresis confirmed the release of genome DNA from HCl- and NaOH-induced LMGs into the culture medium through the trans-membrane tunnel structure (Figure 2d, lanes 2 and 3). The genomic DNA fragments were recovered from the culture supernatants of *Lm* cells treated with HCl and NaOH, respectively (Figure 2d, lanes 2 and 3), compared with the wild-type bacterial cells that showed the complete absence of DNA band (lane 1).

SDS-PAGE analysis exhibited some differences in the remaining proteins extracted from HCl-, H_2_SO_4_-, and NaOH-induced LMGs, respectively (Figure 2e). Specifically, the cells treated with HCl and NaOH showed weaker or missing protein bands when compared with wild-type *Lm* because of the absence of cytoplasmic proteins in these chemically induced LMGs. Numerous studies have suggested that BGs are intact cytoplasm-free due to the loss of cytoplasmic contents [34,50,51]. SDS-PAGE analysis showed similar protein profiles between HCl- and H_2_SO_4_-induced LMGs, but very different protein profiles extracted from NaOH-induced LMGs (Figure 2e). It suggests that acid and alkali have different effects on the bacterial cell envelope. The released proteins from the LMGs were recovered from the culture supernatants of *Lm* cells treated with HCl and NaOH, respectively (Figure 2f, lanes 2 and 3), as compared to wild-type bacterial cells that showed the complete absence of a protein band (Figure 2f, lane 1). Our results supported that the osmotic pressure difference between the cytoplasm and surrounding medium could be the driving force for the rapid release of the cytoplasmic contents, including nucleic acids and cytoplasmic proteins, through the trans-membrane tunnels [34].

### 2.4. Determination of DNA-Free LMGs by Real-Time qPCR

To further confirm the results of agarose gel electrophoresis, we performed real-time qPCR analysis. The results exhibited the normalized reporter-dye fluorescence (dR) as a function of the cycle and indicated that 16S rRNA and *iap* gene fragments were differentially amplified from the three chemical-induced LMGs (Figure 3). In the 16S rRNA, the mean C_t_ values for the treated samples (HCl-, H_2_SO_4_- and NaOH-induced LMGs), positive control (wild-type *Lm*), and negative controls (TE buffer, *E*. *coli*, and *S. enterica* Enteritidis) were 34.07, 36.21, 20.59, 20.36, 40.53, no C_t_ (no cycle of threshold and were therefore not detected) and no C_t_, respectively. In the *iap* gene, the mean C_t_ values for the treated samples (HCl-, H_2_SO_4_-, and NaOH-induced LMGs), positive control (wild-type *Lm*), and negative controls (TE buffer, *E*. *coli* DH5α, and *S. enterica* Enteritidis) were 37.33, 42.28, 22.67, 21.77, no C_t_, 41.08, and no C_t_, respectively. When compared with wild-type *Lm* cells, a complete absence of genomic DNA was observed in HCl- and H_2_SO_4_-induced LMGs, but not in NaOH-induced LMGs. Unlike the results of Figure 2c, detectable DNA concentrations were amplified from NaOH-induced LMGs. Nevertheless, we included NaOH-induced LMGs for further analysis.

Our previous studies showed that NaOH produced complete DNA-free BGs from a Gram-positive bacterium [47] and Gram-negative bacteria [42,45,46]. Similarly, Wu et al. [48] reported that genomic DNA-free LMGs were generated at the MICs of NaOH. These results obtained in the previous studies were in contrast with the results obtained in NaOH-induced LMGs. The difference may not have occurred due to the different cell wall structures of Gram-positive and Gram-negative bacteria. This observation suggested that the acids and alkalis affect bacterial cell walls differently, depending on bacterial species.

### 2.5. Comparison of In Vitro Cytotoxicity for LMGs

Cytotoxicity was compared by the viability of the RAW 264.7 macrophages exposed to HCl- and NaOH-induced LMGs and wild-type *Lm* cells, respectively (Figure 4). *E*. *coli* LPS (5 and 50 μg/mL) showed 88.0 and 60.0% cell viability under the given concentration (bars 2 and 3). Macrophages treated with HCl-induced LMGs (3.7 × 10^6^ CFU/mL) showed the highest cell viability (86.0%, bar 4), while macrophages treated with NaOH-induced LMGs (3.7 × 10^6^ CFU/mL) showed 78.0% cell viability (bar 8). It suggests that HCl-induced LMGs are safer than NaOH-induced LMGs. Regardless of concentration (3.7 × 10^6^, 3.7 × 10^7^, 3.7 × 10^8^, 3.7 × 10^9^ CFU per mL), macrophages exposed to wild-type *Lm* cells showed less than 45.3% cell viability. Therefore, a concentration of 3.7 × 10^6^ CFU/mL for HCl- and NaOH-induced LMGs, and wild-type bacteria was selected for the following cytokine experiments.

### 2.6. In Vitro Study of Cytokine mRNA Expression in Murine Macrophages-Exposed LMGs

Several reports have shown differences in the cytokine production in murine macrophages infected with live and killed *Lm* strains [52,53]. A previous study revealed that killed *Lm* bacteria were much less efficient than live *Lm* bacteria in stimulating TNF-α and IL-1 production, while live and killed bacteria were highly efficient in stimulating IL-6 production [52]. Another study demonstrated that live *Lm* mutants rapidly induced IL-1α, IL-1β, IL-6, and TNF-α mRNAs in murine macrophages, while killed bacteria only induced IL-1β mRNA [53]. Hence, we compared the mRNA induction levels of pro-inflammatory mediators (TNF-α, IL-1β, iNOS, and IFN-γ), anti-inflammatory cytokine (IL-10), and both pro- and anti-inflammatory cytokine (IL-6) in the murine macrophages exposed with PBS buffer, *E*. *coli* LPS, wild-type *Lm* cells, and HCl- and NaOH-induced LMGs, respectively (Figure 5).

Both NaOH-induced and HCl-induced LMGs expressed significantly higher levels of TNF-α mRNA than PBS control in the macrophages at 4 h post-exposure (*p* < 0.001). Besides, a significant difference was observed in NaOH-induced LMGs when compared to HCl-induced LMGs (*p* < 0.01). HCl-induced LMGs expressed significantly higher levels (3.8-fold) of IL-1β mRNA induction than NaOH-induced LMGs in the macrophages at 2 h post-exposure. Both NaOH-induced and HCl-induced LMGs expressed significantly higher levels of IL-6 and IL-10 mRNA levels than PBS control in the macrophages at 4 h post-exposure (*p* < 0.001). However, no significant difference in IL-6 and IL-10 mRNA levels was observed between HCl-induced LMGs and NaOH-induced LMGs. Previously, IL-10 mRNA significantly increased in the macrophages exposed to NaOH-induced *V*. *parahaemolyticus* BGs compared with *E*. *coli* LPS and wild-type *V*. *parahaemolyticus* cells [46]. IL-10 is a main anti-inflammatory cytokine generated during infectious diseases [54], showing a detrimental effect on susceptibility to *Lm* infection [55] and a benefit on the generation of CD8^+^ T cell memory in response to *Lm* infection [56].

NaOH-induced LMGs expressed the highest level of iNOS mRNA in the macrophages at 6 h post exposure (Figure 5). Besides, a significant difference was observed in NaOH-induced LMGs when compared to HCl-induced LMGs (*p* < 0.001). Both HCl-induced and NaOH-induced LMGs expressed significantly higher levels of IFN-γ mRNA than PBS control in the macrophages at 2 h post-exposure (*p* < 0.01 and *p* < 0.001). However, no significant difference in IFN-γ mRNA levels was observed between HCl-induced LMGs and NaOH-induced LMGs [57]. IFN-γ is the main contributor to the cell-mediated immune response via macrophage activation and the increasing antigen presentation via the MHC class I and II pathways. Furthermore, it subsequently activates macrophages to clear intracellular bacteria through reactive oxygen and nitrogen species production [58,59].

Taken together, NaOH-induced LMGs were more efficient than HCl-induced LMGs and wild-type *Lm* bacteria in the induction of TNF-α, IL-10, iNOS, and IFN-γ mRNAs in the RAW 264.7 macrophages, while NaOH-induced LMGs were less efficient than HCl-induced LMGs and wild-type *Lm* bacteria in the induction of IL-1β mRNA. It suggests that NaOH-induced LMGs are the better inducer to activate macrophages to secrete pro-inflammatory mediators except for IL-1β. Additionally, all results revealed that LMGs are different from wild-type *Lm* bacteria in inducing pro-inflammatory mediator mRNAs.

### 2.7. Induction of Humoral Immune Response in Serum and Serum Bactericidal Activity

Serum IgG antibodies significantly increased in BALB/c mice after immunizations and virulent challenges (Figure 6a). At week 2, serum IgG antibody responses in vaccinated groups with HCl-induced LMGs and NaOH-induced LMGs were not significantly different from the PBS buffer-treated control mice (*P* < 0.05). At week 4, both HCL-induced LMGs and NaOH-induced LMGs were significantly higher than the PBS-injected control. At week 8, IgG antibody levels were significantly higher in HCl-induced LMGs (3.7-fold) and NaOH-induced LMGs (3.8-fold) compared with PBS-injected control. Most importantly, serum IgG antibodies in HCl-induced LMGs and NaOH-induced LMGs increased strongly after challenge periods. The bactericidal activities against virulent *Lm* were determined in both the LMG-vaccinated groups and PBS-injected control group (Figure 6b). At week 6, antisera collected from HCl- and NaOH-induced LMGs showed significantly higher bactericidal activity than those of the PBS-injected control group. Although NaOH-induced LMGs induced more IgG antibodies than HCl-induced LMGs, no significant difference was observed between the two groups. Moreover, LMGs vaccinated animals did not show any adverse effects following the inoculation of LMGs.

The data indicated that subcutaneous immunization with LMGs efficiently produced bacteria-killing antibodies, consistent with previous studies in which chemically induced BGs significantly induced serum IgG antibodies throughout the vaccination and challenge periods [42,45,47]. The humoral immune response, in general, has been considered ineffective and unable to protect against *Lm* infection because of the intracellular nature of the bacteria [60]. It suggested that the role of antibodies should be re-evaluated for the protection against *Lm* infection.

### 2.8. Western Blot Analysis of Antibody Response

It would be of great necessity to find the difference of antigenic surface proteins in chemically induced BGs. Western blot analysis showed that antisera collected from HCl- and NaOH-induced LMGs immunized animals reacted similarly to total proteins extracted from *Lm* (Figure 6c). These two antibodies reacted with seven antigenic proteins by respective antibodies. Interestingly, six antigenic proteins (10, 13, 15, 23, 35, and 45 kDa) were common against respective antisera in HCl- and NaOH-induced LMGs immunized animals (Figure 6c). The antigen recognized only by the antiserum against HCl-induced LMGs was a 80 kDa protein, while the antigen recognized only by the antiserum against NaOH-induced LMGs was a 67 kDa protein. These two proteins were of similar sizes to InlA (80 kDa) [61] and InlB (68 kDa) [62]. However, protein bands of sizes corresponding to LAP (104 kDa) [63], ActA (90 kDa) [64], and LLO (58 kDa) [62] were not in the antiserum. It was speculated that these chemicals might degrade relevant antigenic proteins on the bacterial surface [50]. Therefore, we speculated that low molecular weight proteins such as those of 10, 13, 15, 23, 35, and 45 kDa seem to be degraded proteins of high molecular weight proteins. Besides, the InlB recombinant proteins expressed in *E. coli* showed three truncated forms of shorter overlapping peptide fragments and each form yielded functional protein of 23 kDa, 35 kDa, and 45 kDa [2].

### 2.9. Mucosal and Cell-Mediated Immune Responses

To determine the mucosal and cell-mediated immune responses, the cell populations of SigM-B, CD4^+^-T, and CD8^+^-T in immunized groups were examined by FACS analysis (Figure 7). Mice vaccinated with HCl- and NaOH-induced LMGs showed significant differences in SigM-B, CD4^+^-T, and CD8^+^-T cells compared to the PBS-injected mice (*p* < 0.05). The highest percent of CD4^+^-T cell population was found in vaccinated mice with HCl-induced LMGs, while the highest percent of SigM-B and CD8^+^-T cell populations was found in vaccinated mice with NaOH-induced LMGs. The percent of SigM-B cell population was significantly higher in the NaOH-induced LMGs than HCl-induced LMGs, while the percent of CD4^+^-T and CD8^+^-T cell populations was not significantly different between the two LMGs groups. These findings suggests that LMGs induce mucosal and cell-mediated immune responses to *Lm* antigens during virulent challenges.

### 2.10. Protective Efficacy of LMGs against Lm Challenge

The protective efficacy of LMGs was evaluated against virulent challenges. Mice immunized with HCl- and NaOH-induced LMGs, and PBS were challenged with virulent *Lm*. The main sites of *Lm* multiplication are the liver [65,66] and spleen in mice [67]. Therefore, the liver and spleen were obtained at week 2 post-challenge. We observed that the liver and spleen homogenates of the mice vaccinated with HCl- and NaOH-induced LMGs showed significantly reduced bacterial loads compared to the PBS-injected control (Figure 8). Remarkably, subcutaneous vaccination with HCl-induced LMGs and NaOH-induced LMGs efficiently protected against infections. Similarly, subcutaneous immunization of NaOH-induced BGs efficiently produced bactericidal antibodies after virulent challenges [45,47]. Another study reported that anti-ActA and anti-LLO antibody administration significantly reduced bacterial loads in the spleen and liver of *Lm*-infected mice or infected RAW264.7 cells [68]. Under the present experimental condition, HCl-induced LMGs did not cause any clinical signs, while NaOH-induced LMGs caused morphological changes in the spleen and liver (Figure 8).

We performed the Kaplan–Meier survival analysis (Figure 9a) that showed significant differences at inoculation concentrations of *Lm*. At day 11 post-inoculation, the inoculation concentration of 2 × 10^7^ CFU/100 μL showed an average of 33% survival rate, while others (1 × 10^8^, 8 × 10^7^, 6 × 10^7^, and 4 × 10^7^ CFU/100 μL) killed all mice. In Figure 9b, mice immunized with HCl-induced and NaOH-induced LMGs showed much higher survival rates than those injected with PBS buffer during the experimental period. Although the NaOH-induced LMGs showed effective humoral and cell-mediated immune responses and provided good protection against *Lm*, the HCl-induced seem to be a better vaccine candidate due to being completely free of *Lm* DNA and less toxic in vitro and in vivo. In addition, it suggests that immunization with the multi-component vaccine potentially prevents multiple organ infections in contrast with a single component vaccine, which has difficulty generating a strong protective response from a single immunogen [69]. Therefore, our novel strategy of using HCl-induced LMGs as a vaccine candidate could be a new strategy to prevent *Lm* infections.

## 3. Materials and Methods

### 3.1. Bacterial Strain and Culture Condition

Non-living LMGs were produced from *Lm* KCCM 40307 (serotype 1/2a, Korean Culture Center of Microorganisms, Seoul, Korea). The bacterial culture was freshly grown in brain heart infusion (BHI) broth (Becton, Dickinson Company, Franklin Lakes, NJ, USA) at 37 °C in a shaking incubator at 200 rpm. The bacterial cell growth and lysis were monitored by spectrophotometrically measuring the absorbance at 600 nm (OD_600 nm_) (Libra S22, Biochrom Ltd., Cambridge, UK).

### 3.2. Chemicals and Determination of Their MICs

The MICs of HCl, H_2_SO_4_, and NaOH were determined by the conventional twofold broth dilution method [42]. The *Lm* culture was grown in BHI medium and adjusted to a final concentration of 1 × 10^6^ CFU/mL. Four mL of twofold serial dilution of HCl, H_2_SO_4_, or NaOH (stock solution: 50 mg/mL) were added to 1 × 10^6^ CFU/mL of the bacterial culture (4 mL) and incubated at 37 °C for 18 h. The MICs of different chemicals were determined in triplicate. To confirm the MICs, cultures that showed no visible growth were verified by spreading 100 µL of the culture onto LB agar plates. After incubation at 37 °C for 24 h, the CFU/mL was calculated.

### 3.3. Production of LMGs

The LMGs was produced by sponge-like protocol as described previously with some modifications [41,42]. Briefly, the cultured biomass of *Lm* for 72 h was centrifuged at 10,000× *g* for 10 min at 4 °C. The bacterial pellets were collected, washed twice with phosphate-buffered saline (PBS, 5 mM K_2_HPO_4_, 5 mM KH_2_PO_4_, 150 mM NaCl, pH 7.0), and adjusted to a final concentration of 1 × 10^6^ CFU/mL. Stock solutions (1 mL, 5×) of HCl, H_2_SO_4_, and NaOH determined by MICs were added to the bacterial suspension (2 mL). Subsequently, the sterilized distilled water (2 mL) was added to give a final concentration equal to 1× for each chemical. All samples were incubated at 37 °C for 60 min. At different time points (5, 10, 15, 30, 45, and 60 min), the lysis rates of the non-treated control (wild-type) and the chemical-treated bacterial samples were determined by a standard plating procedure. This viability assay for each time point was performed in triplicate. After lysis, chemically induced LMGs were harvested by centrifugation (15 min, 10,000× *g* at room temperature) and washed twice with PBS. The final cell pellets were resuspended in ice-cold PBS and stored at 4 °C until further use.

### 3.4. SEM Analysis

Three chemically induced LMGs (HCl, H_2_SO_4_, and NaOH) were examined for morphological analysis with SEM. Briefly, chemically induced LMG was fixed with 2.5% glutaraldehyde (Millipore Sigma, St. Louis, MO, USA) in PBS at 4 °C for 2 h. After washing with PBS, the samples were fixed in 1% osmium tetroxide at 4 °C for 1 h and subsequently dehydrated in a graded series of ethanol dilutions (30, 50, 70, 90, and 100%). The samples were dried using liquid carbon dioxide, mounted on the holder by silver paint, and sputtered with gold using a polaron high-resolution sputter. Electron micrographs were taken under Hitachi S-4800 Field Emission SEM II (National NanoFab Center, KAIST, Daejeon, Korea) and Carl Zeiss SUPRA55V VP-FESEM (Korea Basic Science Institute, Chuncheon, Gangwon-do, Korea).

### 3.5. Agarose Gel Electrophoresis and SDS-PAGE Analyses

To confirm the absence of genomic DNA in the LMGs, genomic DNA was isolated from three chemically induced LMGs (HCl, H_2_SO_4_, and NaOH), along with control cells, by using a bacterial genomic DNA isolation kit (iNtRON Biotechnology, Seongnam, Gyeonggi-Do, Korea), according to the manufacturer’s instructions. The extracted genomic DNA was analyzed by electrophoresis in 1% agarose gel. To determine the presence of DNA released extracellularly, DNA was concentrated from the culture supernatants using 2 volumes (*v/v*) of ethanol and 1/10 volume (*v/v*) of sodium acetate.

In addition, HCl-, H_2_SO_4_-, and NaOH-induced LMGs and wild-type *Lm* cells were harvested by centrifugation (12,000 rpm for 4 min at room temperature). Respective pellets were washed with PBS, sonicated using a microtip with 20% power for 1 min (repeated 10 times), harvested by centrifugation. The resulting pellets were resuspended in 15 μL of cold PBS, denatured in 5 μL of Laemmli’s loading buffer by boiling for 5 min, and then loaded into 12% sodium dodecyl sulfate-polyacrylamide gel electrophoresis (SDS-PAGE) under constant 40 V for 2 h. In addition, the culture supernatants were precipitated using a 2.5 volume of cold acetone for extracellular protein concentration. The gels were stained in Coomassie brilliant blue solution for 4 h at room temperature and immersed in a destaining solution. The protein concentration was determined using a Bio-Rad Protein Assay Kit (Hercules, CA, USA).

### 3.6. TaqMan Probe-Based Real-Time qPCR

Bacterial cells were treated with the MICs of HCl, H_2_SO_4_, and NaOH, respectively, for 60 min to determine the presence or absence of DNA in the LMGs. Genomic DNA was extracted from various chemically treated LMGs and used as the template for real-time PCR. Based on the NCBI database, we designed primers and probes. The 16S rRNA and *iap* genes of *Lm* were amplified with respective primer sets and probes (Table 2) [70,71]. Real-time PCR was performed in 20 µL HS Prime qPCR Premix with UDG (GeNet Bio, Daejeon, Korea) containing 0.4 µL of each primer (10 pM), 0.4 µL probe (10 pM), 0.4 µL ROXII, and 1 µL DNA (1 ng). Reactions were initiated at 50 °C for 3 min and 95 °C for 10 s, followed by 45 cycles at 95 °C for 5 s and 56 °C for 10 s. Negative controls (TE buffer, *E*. *coli* DH5α, and *S. enterica* Enteritidis), as well as a positive control (wild-type *Lm* cells), were simultaneously included in each run. Each sample was quantified in triplicate and processed three different times with real-time PCR under the same conditions. A relative amount of DNA in the LMGs treated with each chemical was quantified by comparing the *C*_t_ value of each sample to the *C*_t_ value of the TE buffer control. The experiments were analyzed with auto-baseline and manual thresholds chosen from the exponential phase of the PCR amplification. After the data analysis, the *C*_t_ number and the DeltaRn (dRn) were used for statistical analyses. Data were analyzed using Mxpro software (Agilent Technologies, Inc., Santa Clara, CA, USA) and the comparative threshold cycle (2^−ΔΔCt^) method [72].

### 3.7. Assessment of Macrophage-Mediated Cytotoxicity

Murine macrophage cells (KCLB:40071 and RAW 264.7) were purchased from the Korean Cell Line Bank (Seoul, Korea) and cultured in BD Falcon™ 96-well plates (BD Bioscience, Franklin, NJ, USA) for 24 h at 37 °C and 5% CO_2_. The cells (5 × 10^4^ cells/well) were then treated with various concentrations (3.7 × 10^6^, 3.7 × 10^7^, 3.7 × 10^8^, 3.7 × 10^9^ CFU/mL) of wild-type *Lm* cells, HCl- and NaOH-induced LMGs, respectively, and incubated for a further 24 h. PBS- and LPS-treated (5 and 50 μg/mL; Millipore Sigma) macrophages were used as negative and positive controls, respectively. The cell density was then assessed by using a Cell Counting Kit-8 (CCK-8; Millipore Sigma). Absorbance was measured at 450 nm, and all experiments were performed in triplicate. Cytotoxic activity is expressed as the percentage of cell viability using the following formula: % cytotoxicity = (1 − A_450 nm_ of target cells/A_450 nm_ of control cells) × 100 [73].

### 3.8. Quantitative Analysis of Cytokine mRNA by Reverse Transcription (RT)-qPCR

RAW 264.7 cells (1.0 × 10^6^ cells/well) were cultured in 24-well flat-bottom plates and treated with the HCl- and NaOH-induced LMGs and wild-type *Lm* cells, respectively, at a concentration of 3.7 × 10^6^ CFU/mL. After 2, 4, and 6 h stimulation, total RNA was isolated from the respective macrophages by using RNAiso Plus (Takara Bio, Shiga, Japan), according to the manufacturer’s instructions. Expression levels of mRNA for tumor necrosis factor (TNF)-α, interleukin (IL)-1β, inducible nitric oxide synthase (iNOS), interferon (IFN)-γ, IL-6, and IL-10 were quantified by RT-qPCR amplification. Sequences for the primers of target genes are listed in Table 2. The RT reaction was performed in a 20 μL reaction mixture containing 300 ng of total RNA, 50 mM Tris-HCl (pH 8.3), 75 mM KCl, 8 mM MgCl_2_, 10 mM DTT, 0.1% NP-40, 40 mM dNTP, 2 pM of respective primer set, 20 U of RNase inhibitor, and 200 U PrimeScript Reverse Transcriptase (Takara Bio). The thermal cycler for RT was programmed as follows: 1 cycle at 50 °C for 30 min and 70 °C for 15 min. The resulting cDNA was amplified in a 20 μL reaction mixture containing 10 μL 2× SYBR^®^ Premix Ex Taq™ II (Tli RNaseH Plus; Takara Bio), 0.2 μL ROX reference dye II, 0.4 μL of 10 μM each forward and reverse primer (Table 2), and 1 ng of cDNA. The Stratagene Mx3005P cycler (Agilent Technologies, Inc.) was programmed as follows: 1 cycle at 95 °C for 30 s, 30 cycles of denaturation at 95 °C for 5 s, and primer annealing and extension at 60 °C for 34 s. All genes were amplified in triplicate, and the differences in cDNA concentration normalized to glyceraldehyde 3-phosphate dehydrogenase (GAPDH).

### 3.9. Experimental Animals, Vaccination, and Challenge

BALB/c mice were maintained at 23–25 °C in a 12 h light/dark cycle and had access to a standard pellet diet and water ad libitum. All experimental procedures were approved by the institutional animal care and use committee in Pai Chai University (PCU-20163). Twenty-four male BALB/c mice (five-week-old) were equally divided into three groups (n = 8 per group) and injected subcutaneously with 100 µL of sterile PBS (non-vaccinated control), HCl-induced LMGs (1 × 10^7^ CFU/mL), and NaOH-induced LMGs (1 × 10^7^ CFU/mL), respectively. Animals from all three groups were immunized three times at two-week intervals (week 1, week 3, and week 5). Two weeks after the last immunization (week 7), all animals were challenged orally with 2 × 10^7^ CFU/ 100 μL of virulent *Lm* in PBS. To determine the immune response, blood samples were taken from tail veins of individual mice at two-week intervals during immunizations and after challenges.

### 3.10. Measurement of Antibody Response by ELISA

Sera from vaccinated and non-vaccinated control BALB/c mice were determined for the presence of specific immunoglobulin G (IgG) by indirect enzyme-linked immunosorbent assay (ELISA). Briefly, microtiter plates were coated with 100 µL of antigen (1 × 10^6^ cells/mL) in coating buffer (pH 9.6) and incubated for 2 h at room temperature. The plates were then washed three times with PBS containing Tween-20 (PBS-T) and then blocked with 1% bovine serum albumin (BSA) in PBS-T for 2 h at room temperature. After washing, 100 µL of serially diluted sera were added and incubated for 2 h at room temperature. Plates were washed three times with PBS-T, and then 100 µL of goat anti-rat IgG conjugated alkaline phosphatase (1:30,000; Millipore Sigma) in PBS-T with 1% BSA were added and incubated for 2 h at room temperature. Plates were then washed three times with PBS-T, and the color was developed using 100 µL of p-nitrophenyl phosphate substrate (Millipore Sigma) and incubated for 30 min at room temperature in the dark. The reaction was stopped by adding 100 µL of 3M NaOH, and plates were read at 405 nm with iMark Microplate Reader (Bio-Rad).

### 3.11. Western Blot Analysis of Cell Envelope Proteins

*Lm* cell envelope proteins were extracted as described in Section 3.5. Twenty μg of protein were loaded and subjected to 12% SDS-PAGE. Western blots were probed with HCl- and NaOH-LMGs vaccinated or PBS-injected antisera (6 weeks, 1:1000) in TBS-T buffer containing 5% non-fat dry milk as primary antibodies. Goat anti-mouse IgG conjugated alkaline phosphatase (1:7000; Millipore Sigma) was used as a secondary antibody. Detection was developed using NBT Substrate Powder containing nitroblue tetrazolium and 5-bromo-4-chloro-3-indolylphosphate (Thermo Fisher Scientific, Waltham, MA, USA).

### 3.12. Flow Cytometric Analysis

B- and T-cell populations were examined by collecting blood samples from immunized and non-immunized control rats on the seventh day (week 6) after the final immunization. Peripheral blood mononuclear cells (PBMC) were prepared by using Histopaque-1077 (Millipore Sigma) according to the manufacturer’s protocol. PBMCs were washed three times with cold PBS and stained with appropriately diluted Mouse T lymphocyte Subset Antibody Cocktail (PE-Cy^TM^ 7 CD3e, PE CD4, and FITC CD8) purchased from BD Biosciences (San Jose, CA, USA) at 4 °C for 30 min in the dark. After incubation, all samples were washed three times with PBS and resuspended in 0.5 mL of PBS. Data were collected with fluorescence-activated cell sorter (FACS) analysis using a BD FACSCanto II flow cytometer and BD FACSDiva software (BD Biosciences).

### 3.13. Determination of Serum Bactericidal Activity and Bacteriological Analysis

Serum bactericidal activity (SBA) was performed as follows. In brief, 100 µL of bacterial suspension (1 × 10^6^ CFU/mL) were added to 25 µL of serum and incubated at room temperature for 1 h. After incubation, the mixture was spread onto BHI (Becton, Dickinson Company) agar media and incubated at 37 °C for 48 h. For a control, serum was replaced by PBS. The SBA percent was determined as follows: SBA = {1—(the number of viable bacteria after serum treatment/the number of viable bacteria after PBS treatment)} × 100%. All experimental animals were sacrificed at two weeks (week 9) post-challenge to evaluate the protective efficacy of the LMG vaccine. The internal organs such as the liver and spleen were collected and aseptically homogenized in 5 mL of sterilized cold PBS using T10 basic ULTRA-TURRAX Homogenizer (Ika-Werke GmbH & Co., Staufen, Germany). Tenfold serial dilutions of these tissue homogenates were plated onto the selective media, and the numbers of CFUs were determined.

### 3.14. Statistical Analyses

Data were analyzed for statistical significance with one-way ANOVA by the SPSS software (version 21.0, SPSS Inc., Chicago, IL, USA). All data were expressed as mean ± standard error of the mean and compared using Duncan’s multiple range tests. Differences were considered statistically significant if *p* < 0.05. Kaplan-Meier survival curve was constructed using GraphPad Prism 5.0 (GraphPad Software, San Diego, CA, USA), and the analysis was done using Log-rank (Mantel-Cox) test. Results were considered statistically significant if *p* < 0.05.

## 4. Conclusions

In conclusion, non-living LMGs have been successfully generated at MICs of HCl and NaOH, and the approach is a rapid and cost-effective method compared to other known methods [28,29,30,31,32,33,34,35,36,37,38,39,40,50,51]. Most importantly, we have shown that immunization with LMGs induced significant humoral and cell-mediated immune responses and protected strongly against virulent challenge in BALB/c mice. Therefore, our present findings could be helpful in future vaccine developments against *Lm* infection.

## Figures and Tables

**Figure 1 ijms-23-01946-f001:**
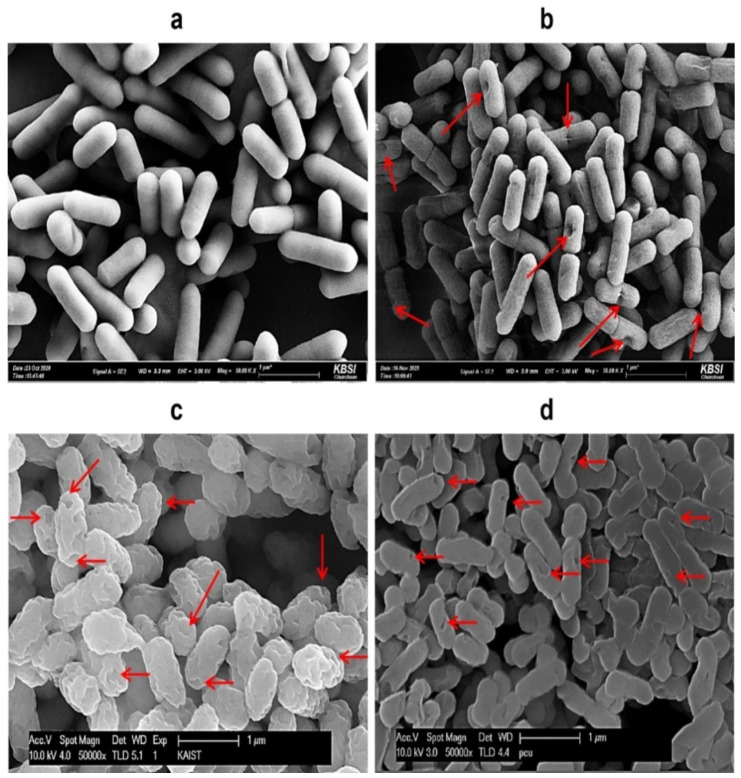
Comparison of SEM images of (**a**) wild-type *L. monocytogenes* cells; (**b**) HCl-induced LMGs; (**c**) H_2_SO_4_-induced LMGs; and (**d**) NaOH-induced LMGs. Small arrows show the trans-membrane lysis pores. Electron micrographs were taken by Hitachi S-4800 Field Emission SEM II (**a,b**) and Carl Zeiss SUPRA55V VP-FESEM (**c**,**d**).

**Figure 2 ijms-23-01946-f002:**
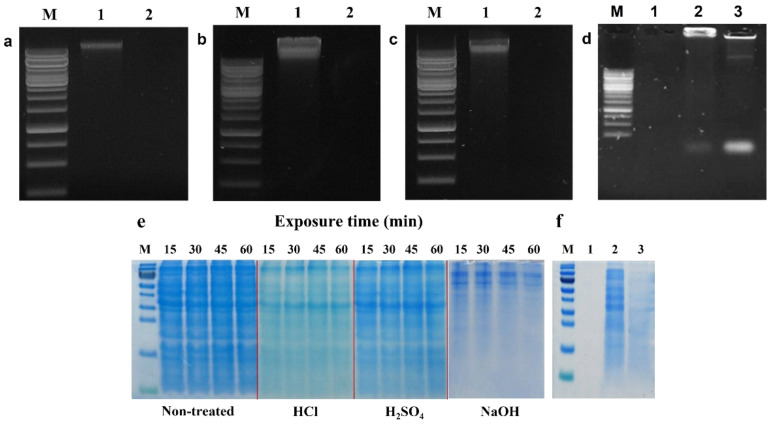
Agarose gel (1%) electrophoresis of genomic DNAs extracted from LMGs chemically induced by (**a**) HCl; (**b**) H_2_SO_4_; (**c**) NaOH (lane 1, wild-type *L. monocytogenes*; and lane 2, chemically induced LMGs); and (**d**) concentrated DNA from culture supernatants of wild-type *L. monocytogenes* (lane 1); HCl- (lane 2); and NaOH-induced LMGs (lane 3). M, 1 kb Marker Ladder. (**e**) During LMGs production, proteins were extracted from wild-type *L. monocytogenes*, and HCl-, H_2_SO_4_-, and NaOH-induced LMGs at 15, 30, 45, and 60 min. Each lane was loaded with 10 μg of proteins. M: Protein Marker. (**f**) Remaining proteins in culture supernatants from wild-type *L. monocytogenes* (lane 1); HCl- (lane 2); and NaOH-induced LMGs (lane 3). Except for lane 1 (0.15 μg), lanes 2 and 3 were loaded with 2.7 μg of proteins.

**Figure 3 ijms-23-01946-f003:**
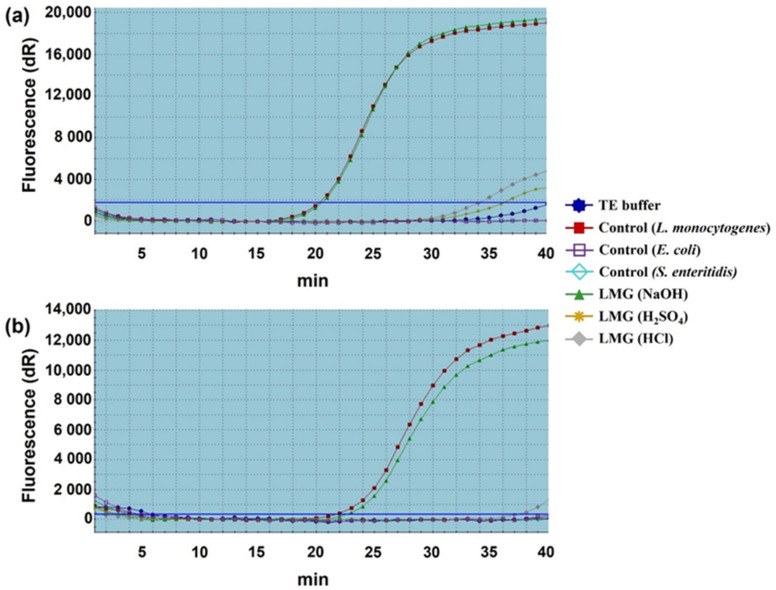
Quantitative analysis of remnant DNA extracted from chemically induced LMGs. (**a**) 16 s rRNA and (**b**) *iap* genes amplified with TaqMan probe-based real-time qPCR. The DNA quantity of respective LMGs was compared with wild-type *L. monocytogenes* cells (positive control); and TE-buffer, *E. coli* DH5α, and *S. enterica* Enteritidis (negative controls).

**Figure 4 ijms-23-01946-f004:**
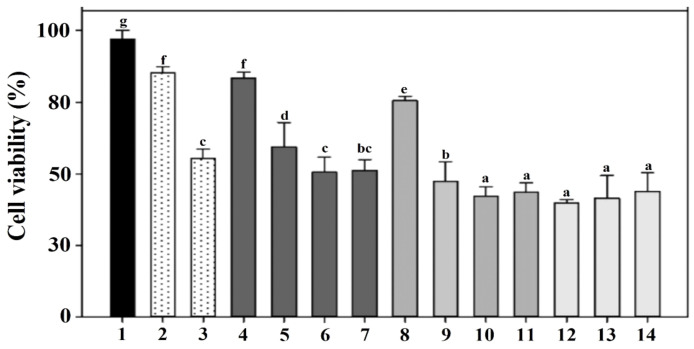
To determine cytotoxicity, murine macrophages RAW 264.7 were exposed to PBS buffer (bar 1); 5 and 50 μg/mL LPS from *Escherichia coli* (bars 2–3); 3.7 × 10^6^, 3.7 × 10^7^, 3.7 × 10^8^, 3.7 × 10^9^ CFU per mL of HCl-induced LMGs (bars 4–7); NaOH-induced LMGs (bars 8–11); and 3.7 × 10^6^, 3.7 × 10^7^, 3.7 × 10^8^ CFU per mL of wild-type *L. monocytogenes* (bars 12–14), respectively. Macrophages at 24 h post-exposure were collected for analysis of cell viability using a Cell Counting Kit-8. Absorbance was measured at 450 nm, and all experiments were performed in triplicate. All data are expressed as the mean ± the standard error of the mean. The letters above bars represent a significant difference between treatments (*p*
*<* 0.05).

**Figure 5 ijms-23-01946-f005:**
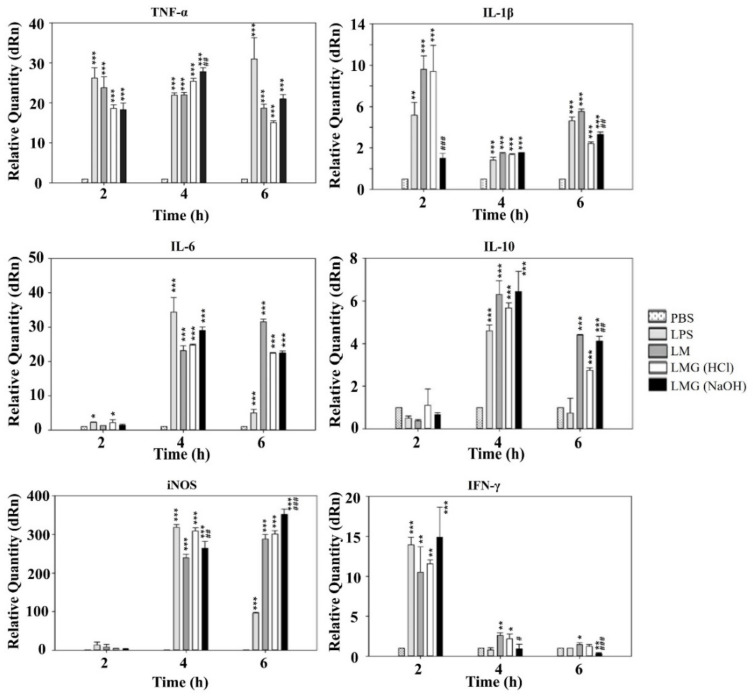
LMGs-exposed murine macrophage (RAW 264.7) stimulates both pro- and anti-inflammatory cytokine production. At 2–6 h post-exposure with LPS from *E. coli*, *L. monocytogenes* wild-type cells (LM), HCl-induced LMGs, and NaOH-induced LMGs, each macrophage was collected for analysis of gene expression for cytokines of TNF-α, IL-1β, iNOS, IL-6, and IL-10 by RT-qPCR. Data are representative of three independent experiments with each sample run in triplicate. All data are expressed as the mean ± the standard error of the mean. *** *p* < 0.001, ** *p* < 0.01, and * *p* < 0.05 indicate a significant difference from a negative control (PBS buffer). ^###^ *p* < 0.001, ^##^ *p* < 0.01, and ^#^ *p* < 0.05 indicate a significant difference between HCl-induced LMGs and NaOH-induced LMGs.

**Figure 6 ijms-23-01946-f006:**
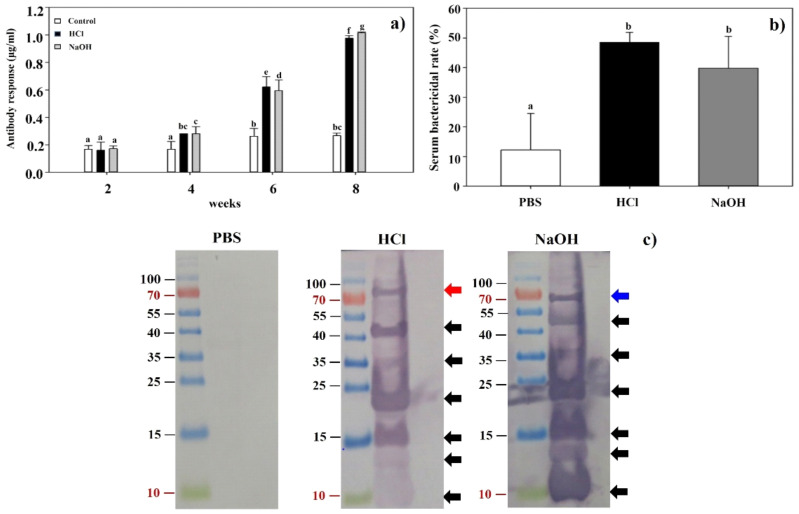
(**a**) The levels of IgG antibody in BALB/c mice vaccinated with LMGs were determined by indirect ELISA using *Lm* as an antigen. Antisera were obtained from mice in three groups (PBS-treated control, HCl-induced LMGs and NaOH-induced LMGs) at 2, 4, 6, and 8 weeks and used as an antibody. Different letters (a, b, bc, c, d, f, and g) indicate significant differences between the antibody responses of the immunized (HCl and NaOH) and non-immunized (PBS) groups (*p* < 0.05). (**b**) Serum bactericidal activities of BALB/c mice subcutaneously injected with PBS, HCl-induced LMGs, and NaOH-induced LMGs, respectively. The letters (a and b) indicate significant differences between serum bactericidal activities of the immunized (HCl and NaOH) and non-immunized (PBS) groups (*p* < 0.05). (**c**) Western blot analysis of total proteins extracted from *Lm* cells that were probed with antiserum from BALB/c mouse injected with PBS, HCl-induced LMGs, and NaOH-induced LMGs, respectively. Data were expressed as means ± standard errors of the means.

**Figure 7 ijms-23-01946-f007:**
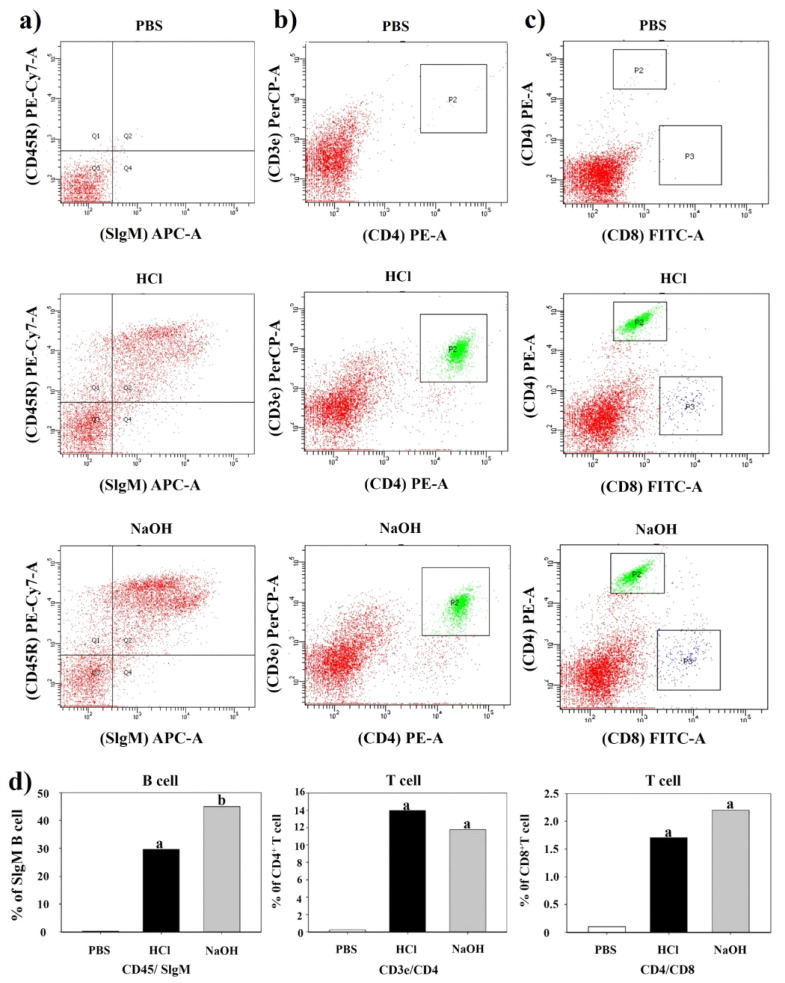
Assessment of B cells, CD4^+^ and CD8^+^ T-cells by FACS analysis. Populations of B cells, CD4^+^, and CD8^+^ T-cells from PBS-injected control, HCl-induced LMGs immunized, and NaOH-induced LMGs immunized groups one week (week 6) after the final immunization. Red color areas represent reaction intensities between CD45R marker and SIgM (a column), between CD3e marker and CD4 (b column), and between CD4e marker and CD8 (c column). P2 box indicates a strong reaction between CD3 marker and CD4 (b column). P2 box and P3 box indicate CD4 reaction and CD 8 reaction, respectively (**c**) column. (**d**) row Percent (%) population of B cells, CD4^+^ T cells, and CD8^+^ T cells corresponding (**a**–**c**) analyses.

**Figure 8 ijms-23-01946-f008:**
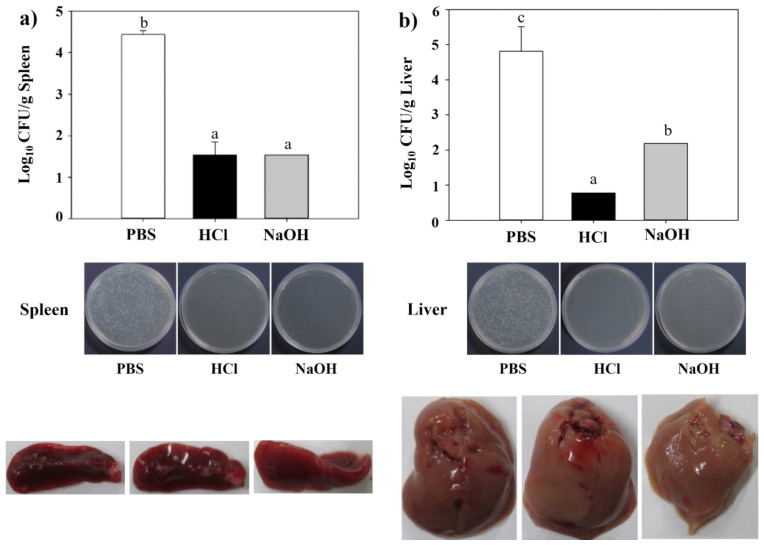
Bacterial loads, viable cell counts and morphological changes in spleen (**a**) and liver (**b**) homogenates after a virulent *Lm* challenge. All data were expressed as mean ± standard error of the mean and compared using Duncan’s multiple range tests. The letters (a–c) indicate significant differences between the bacterial burden of the immunized (HCl and NaOH) and non-immunized groups (PBS).

**Figure 9 ijms-23-01946-f009:**
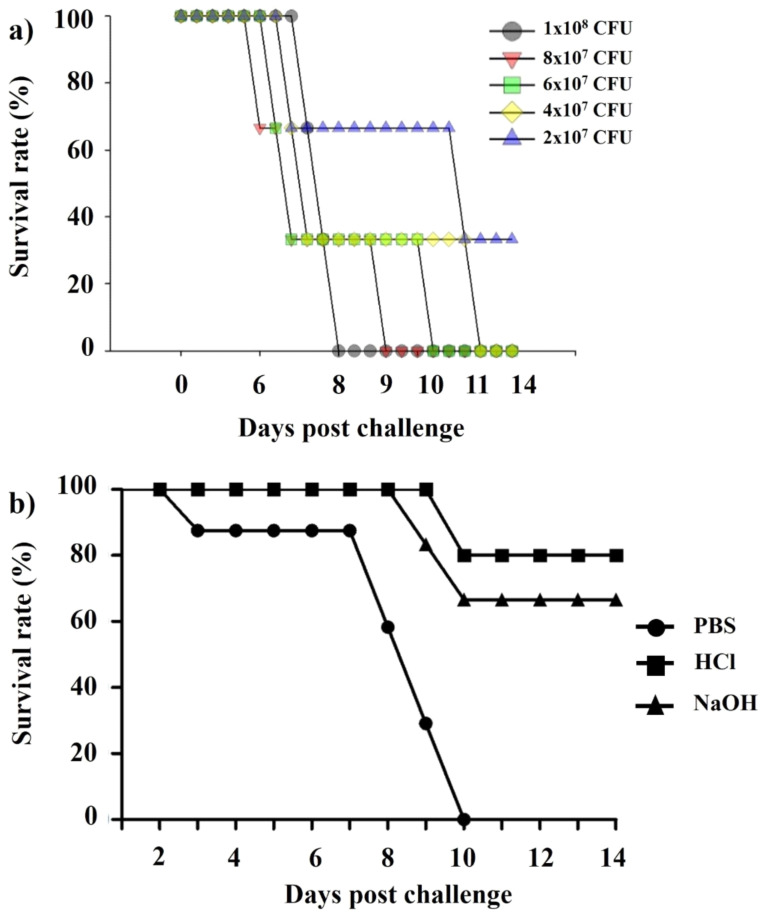
(**a**) Kaplan–Meier survival analysis for the BALB/c mice (*n* = 5) inoculated with different concentrations (1 × 10^8^, 8 × 10^7^, 6 × 10^7^, 4 × 10^7^, and 2 × 10^7^ CFU /100 μL) of *Lm* and (**b**) Kaplan–Meier survival analysis for BALB/c mice (*n* = 8) immunized with PBS, HCl-induced LMGs, and NaOH-induced LMGs and challenged subcutaneously with *Lm* (2 × 10^7^ CFU /100 μL). All experiments were performed twice. Kaplan–Meier survival analysis showed significant differences between immunized (HCl and NaOH-induced LMGs) and non-immunized control (PBS) by the Mantel–Cox log-rank test (*p* < 0.01).

**Table 1 ijms-23-01946-t001:** Minimum inhibitory concentrations (MICs) and minimum bactericidal concentrations (MBC) of chemicals treated in *L. monocytogenes* culture.

Chemicals	MIC (mg/mL)	MBC (mg/mL)
Hydrochloric Acid	6.25	12.5
Sulfuric Acid	12.5	25.0
Sodium Hydroxide	6.25	12.5

**Table 2 ijms-23-01946-t002:** Primer or probe sequences of the targeted genes in real-time qPCR.

Target Gene	Primer	Sequences (5′–3′)	Method	Reference
16s rRNA	L16Sq-F ^1^	GGAATTCCACGTGTAGCGGTGAAAT	Taqman probe	This study
	L16Sq-R	GACTACCAGGGTATCTAATCCTGTTTG		
	L16Sq-TM	AACACCAGTGGCGAAGGCGA		
iap	LMIAPq-F	GTGCAAGTGCTATTATTGCTGAA	Taqman probe	This study
	LMIAPq-R	AGATTGTACGTGGAAGGGAGATA		
	LIAPq-TM	ATGTAGTTGGTCCGTTACCACCC		
TNF-α	TNq-F	ATGAGCACAGAAAGCATGATCCG	SYBR Green	This study
	TNq-R	GCTGAGACATAGGCACCGC		
Il-1β	IL1q-F	ATGGCAACTGTTCCTGAACTCAACT	SYBR Green	This study
	IL1q-R	AGTAGCCCTTCATCTTTTGGGG		
IL-6	IL6q-F	ATGAAGTTCCTCTCTGCAAGAGACT	SYBR Green	This study
	IL6q-R	GTCTCCTCTCCGGACTTGTGA		
IL-10	IL10q-F	ATGCCTGGCTCAGCACTGCTA	SYBR Green	This study
	IL10q-R	CTGGGAAGTGGGTGCAGTTATTG		
IL-12	IL12q-F	ATGTGTCAATCACGCTACCTCCT	SYBR Green	This study
	IL12q-R	GACTGGCTAAGACACCTGGC		
iNOS	NOq-F	ATGAACCCCAAGAGTTTGACCAGA	SYBR Green	This study
	NOq-R	GGAGCCATAATACTGGTTGATGAAC		
GAPDH	GHq-F	ATGGTGAAGGTCGGTGTGAACG	SYBR Green	This study
	GHq-R	CAATGAAGGGGTCGTTGATGGC		

^1^ F: forward, R: reverse.

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
