# Peer review of "Protective Immunity against Listeria monocytogenes in Rats, Provided by HCl- and NaOH-Induced Listeria monocytogenes Bacterial Ghosts (LMGs) as Vaccine Candidates"

_ijms, 2022, doi:10.3390/ijms23041946_

Round 1

Reviewer 1 Report

The authors investigated the “Protective immunity against Listeria monocytogenes in rats, provided by HCl- and NaOH-induced Listeria bacterial ghosts (LMGs) as vaccine candidates”. The topic is novel and provides deep insights into Listeria bacterial ghosts (LMGs) as vaccine candidates. 

This article is well organized and discussed the undermined aspects of LMGs and their applications in immunology. However, authors need to address the following concerns before publication.

  1. Few suggestions include, Extensive English language revisions, expand introduction, write a strong conclusion. 
  2. The authors haven’t stated the potential clinical side effects of LMGs (if any) post-administration in vivo and in vitro. 
  3. The cell and tissue uptake percentage and assimilations of various LMGs need to be more discussed.
  4. Some references are not related to the subject matter in the text as Ref. 48.
  5. In the cytotoxicity (Page 6, Line 185): correct the figure number from figure 2 to figure 4.
  6. The authors should add the analysis of variance test between groups of pro-inflammatory mediators and anti-inflammatory cytokines for NaOH/HCl induced LMGs.
  7. The chemical method used for production of LMGs is a modified method of sponge-like protocol. The authors need to document this method.
  8. The ethical approval number should be mentioned.
  9. The strategy may open the door to produce BGs from Gram-positive bacteria. Remove this statement. BGs from Gram-positive bacteria has been prepared previously.

Author Response

Manuscript ID: ijms-1447112

Title: Protective immunity against Listeria monocytogenes in rats, provided by HCl- and NaOH-induced Listeria bacterial ghosts (LMGs) as vaccine candidates

Reviewer 1

Comments and Suggestions for Authors

The authors investigated the “Protective immunity against Listeria monocytogenes in rats, provided by HCl- and NaOH-induced Listeria bacterial ghosts (LMGs) as vaccine candidates”. The topic is novel and provides deep insights into Listeria bacterial ghosts (LMGs) as vaccine candidates. 

This article is well organized and discussed the undermined aspects of LMGs and their applications in immunology. However, authors need to address the following concerns before publication.

1. Few suggestions include, Extensive English language revisions, expand introduction, write a strong conclusion. 

Response: Thank you for your valuable suggestions. We have revised manuscript as per reviewer’s suggestion.

2. The authors haven’t stated the potential clinical side effects of LMGs (if any) post-administration in vivo and in vitro. 

Response:  Thank you for your valuable suggestions. In vitro assay, we have done cytotoxicity assay using RAW 264.7 cells and described the results in lines 189-199. It is difficult to observe any clinical side effects at RAW 264.7 cells. In vivo assay, we observed clinical side effects in the morphology of spleens and livers in mice treated with NaOH-induced LMGs compared to PBS and HCl-induced LMGs and described the results in lines 380-383. However, we didn’t analyze any biochemical markers to analyze the clinical side effects and try this for a future study.

3. The cell and tissue uptake percentage and assimilations of various LMGs need to be more discussed.

Response: It’s good suggestion though we didn’t investigate LMGs uptake into cells and tissues. Therefore, we can’t discuss about this and will consider this in a future study. Thank you for your idea.

4. Some references are not related to the subject matter in the text as Ref. 48.

Response: We really appreciate the reviewer’s effort for pointing out this. We have removed those references in the revised manuscript.

5. In the cytotoxicity (Page 6, Line 185): correct the figure number from figure 2 to figure 4.

Response: Thank you for notifying us. We have corrected the figure number in the revised manuscript.

6. The authors should add the analysis of variance test between groups of pro-inflammatory mediators and anti-inflammatory cytokines for NaOH/HCl induced LMGs.

Response: Thank you for indicating us. We have revised extensively to simplify the results and described statistical difference between HCl-induced LMGs and NaOH-induced LMGs. Besides, we have analyzed cytokine mRNA expression using P values and changed to a new Figure 5 and legend.

7. The chemical method used for production of LMGs is a modified method of sponge-like protocol. The authors need to document this method.

Response: Thank you for the recommendation. We have revised the manuscript according to the reviewer’s suggestion.

8. The ethical approval number should be mentioned.

Response: Thank you for the suggestion. We have added ethical approval number in the revised manuscript.

8. The strategy may open the door to produce BGs from Gram-positive bacteria. Remove this statement. BGs from Gram-positive bacteria has been prepared previously.

Response: Thank you for indicating us. We have removed the statement and revised the  manuscript as per reviewer’s suggestion.

Reviewer 2 Report

Ji et al. generated Listeria monocytogenes bacterial ghosts (LMGs) using different acidic and alkaline chemicals (HCl, H2SO4 and NaOH) and demonstrated that these LMGs could elicit immune responses and protect mice against L. monocytogenes infection. This study is well designed and the potential of LMGs as vaccine was thoroughly investigated using various methods. However, the current manuscript would benefit from further revisions that are suggested below.

  1. Although chemically generated LMGs are heavily examined in this study, it is not clear why this investigation is necessary, i.e., what novelty this study can bring. My understanding is that NaOH was already employed to generate BGs from L. monocytogenes, which successfully stimulated immune response and protected against L. monocytogenes. Then, why is generating BGs with acids important if the alkaline procedure is already working fine?
  2. Discussion is rather weak and should be carefully revised. The most obvious weakness is that, while many prior studies were described in Results and Discussion, it was often not clear why they were described and how they were related to the data. Specific examples are as follows:

          Lines 233-237 and 240-242: I cannot understand what purpose these statements serve here. What information do the authors want to convey via these statements?

          Lines 283-290: It is not clear how these statements are related to the data obtained in this study.

          Lines 338-346: How is this part related to the data?

  1. Lines 215-229: This part should be further streamlined to convey what these results indicate and why they are important.
  2. Although LMGs were created with three different chemicals, i.e., HCl, H2SO4 and NaOH, all the immunological and animal experiments were conducted with HCl- and NaOH-induced LMGs. Why were these selected for those experiments?
  3. Why was strain KCCM 40307 used in this study instead of the extensively studied lab strains? Which CC/ST does this strain belong to? Is the whole genome sequence available for this strain?
  4. More details should be provided for experimental procedures. For instance, manufacturer information lacks for multiple chemicals and equipment, and volumes of substances that were administered to mice are not shown as well. Also, in line 553, what exactly does "selective media" mean?
  5. Unclear statements abound and need to be edited to improve clarity. See specific comments.
  6. Cite figures and tables more frequently. See specific comments for some examples.
  7. Provide references whenever necessary. Take a look at specific comments for details.
  8. Figures and their legends need to be re-worked. See specific comments.
  9. Provide strain names for E. coli, and S. enteritidis. Show the full name for "S. enteritidis". If this pertains to Salmonella, Change "S. enteritidis" to "S. Enteritidis" with "Enteritidis" unitalicized since "Enteritidis" is a serovar. Also, "S. enteriditis" seems to be a typo for "S. enteritidis". 

Specific comments:

Abstract

          Lines 17 and 18: Revise the statement since "observed" seems to be redundant with "proved".

          Lines 21-23: Show what this statement indicates.

          Line 26: Change "bactericidal activities" to "bactericidal activities of serum".

Introduction

          Line 38: Change "gram-positive" to "Gram-positive".

          Line 39: Change "pathogens of food-borne infection" to "foodborne pathogens".

          Line 40: Change "at the cell envelope" to "in the cell envelope".

          Lines 43-46: Revise this sentence to improve clarity.

          Line 48: Change "a host cell receptor known as" to "the host cell receptor".

          Lines 48-50: Why are these four proteins selected among many virulence factors involved in L. monocytogenes infection? Any particular reason? Also, I recommend changing "such as" to, "i.e.," since "such as" indicates that the listed proteins are a portion of four virulence factors, which is not the case here. Provide references.

          Line 54: InlB has GW repeats but does not harbor LPXTG motif. Take a look at 2009 paper by Bonazzi et al. titled "Listeria monocytogenes Internalin and E-cadherin: From Bench to Bedside."

          Lines 60 and 61: Provide additional references for hypervirulent strains since references 18 and 19 only pertain to antibiotic resistance.

          Lines 61-63: Provide references.

          Lines 63 and 230: Change "triple-virulence-genes deletion mutant" to "triple-virulence-gene deletion mutant".

          Line 66: Change "deliver" and "bind" to past tense.

          Line 67: Change "platform as" to "platform for".

          Line 74: Change "forming" to "possessing".

          Lines 75 and 76: Change "retain" and "lacking" to "retained" and "lacked", respectively.

          Line 81: Clarify "100% lysis rate". Does this mean the state of being completely lysed?

          Line 82: Provide further information for "potential risks".

          Lines 83 and 84: Revise "sodium ... (MIC)" for clarity. Also, the full name for MIC is "minimum inhibitory concentration".

          Lines 88 and 89: Change "strongly bactericidal effects" to "strong bactericidal effects". Change "are applied" to past tense and provide references.

          Lines 89-91: This statement needs to be reworked for clarity.

          Lines 91 and 92: Revise "whether BGs ... alkali treatment" for clarity.

          Line 98: "some" sounds vague; revise it.

Results and Discussion

          Line 120: Double-check whether Figure 2e is suitable here since figure 2e shows proteins.

          Lines 121-123: Cite relevant figures.

          Line 128: Change "HCl-" to "HCl".

          Line 129: Why do the authors think that the protein profile is due to "absence of cytoplasmic proteins"? Provide a further explanation and, if available, relevant references.

          Lines 130-132: Change "extracted from" to "between". Cite relevant figures.

          Line 132: Change "suggested" to "suggests".

          Line 135: Change "showed" to "that showed".

          Line 160: Change "are 34.07" to "were 34.07". Revise other similar instances accordingly.

          Line 161: Elaborate "no Ct".

          Line 171: Change "based on" to "at".

          Lines 172 and 173: This statement could be further abbreviated. 

          Lines 173-175: This statement is rather redundant and should be placed in the prior paragraph. Cite figures instead of "As shown in Figures 3a and 3b".

          Line 184: Change "The cytotoxicity" to "Cytotoxicity".

          Line 186: Please show why E. coli LPS was used in this assay.

          Line 189: Change "It suggested" to "It suggests".

          Line 190: Show the concentrations tested here.

          Lines 191-192, 210, and 317: Change "In this respect"; this transition word does not sound natural.

          Lines 203 and 204: Provide references for "Some reports". I recommend changing "Some" to "Several".

          Lines 215-218: Cite relevant figures and remove "In this study".

          Lines 221 and 223: Remove "Among treatments".

          Lines 230 and 231: Clarify whether "vaccine strain" is L. monocytogenes or not. If so, which strain?

          Line 233: Change "group. [20]." to "group [20]."

          Lines 238 and 239: Cite figures and change "post-exposure" to "post exposure".

          Lines 245-247: Provide references.

          Line 252: Change "suggested" to "suggests".

          Line 255: Change "to induce" to "in inducing".

          Lines 265 and 266: I recommend citing Figure 6a rather than Figure 6.

          Line 266: Change "On week 2" to "At week 2" and revise other similar instances as well.

          Line 269: Clarify "significantly high". Higher than what?

          Line 270: Change "3.7 fold" to "3.7-fold" and revise similar instances as well.

          Lines 273 and 274: Re-write the statement "Antisera ... virulent Lm" to improve clarity.

          Lines 283-285: Provide references.

          Line 303: Change "a great necessity" to "of great necessity".

          Lines 304 and 305: Revise the statement for clarity.

          Lines 305-307: Clarify "antisera ... respectively" and cite figures.

          Lines 308 and 309: Revise the statement for clarity. Also, it would be helpful if the authors show the size of six antigenic proteins commonly found in two LMGs.

          Line 311: Change "80 kDa protein" to "a 80 kDa protein" and revise a similar instance in line 312.

          Line 313: Change "showed similar sizes as" to "were of similar sizes to". Provide references for InlA and InlB.

          Lines 313 and 314: Change "corresponding sizes of" to "of sizes corresponding to". Provide references for LAP, ActA, and LLO.

          Lines 315 and 316: Change "It is speculated" to "It was speculated". Revise "some relevant antigenic proteins" since it sounds vague.

          Line 317: Change "such as" to "such as those of".

          Lines 317-321: Revise it for clarity.

          Line 322: Change "Escherichia coli" to "E. coli". 

          Lines 323-325: I do not think that the detailed information on truncated InlB fragments such as "F3 fragment = cap + LRR" is necessary.

          Line 327: Provide a reason for FACS analysis instead of "In the present study".

          Lines 333 and 334: In which was the percent of SigM-B cell population higher? Be more specific in describing the data.

          Lines 335 and 336: Change "suggested" to "suggests".

          Lines 339 and 340: Revise "phagocytosis-mediated ... mucosa" to improve clarity.

          Line 341: Change "into a cytoplasm" to "into cytoplasm".

          Line 353: Remove "In this study".

          Line 356: Revise "at two weeks post-challenge" to improve clarity.

          Line 357: Clarify "vaccinated with ... LMGs".

          Line 362: Change "Another finding" and "anti-ActA and anti-LLO antibodies administration" to "Another study" and "anti-ActA and anti-LLO antibody administration".

          Lines 371-378: This part should be placed at the beginning of section 2.10, i.e., line 353.

          Lines 372 and 373: Clarify "showed ... concentrations of Lm".

          Line 373: Change "At 11 days post-inoculation" to "At day 11 post inoculation".

          Line 374: Provide specific concentrations tested here after "others".

          Lines 375 and 376: This statement is vague and will confuse readers. What data are presented in Figure 9b? "Significant differences" in what? If "significant differences" are present, which group showed a higher value?

          Lines 376-378: Move this statement to the next paragraph. Also, "concluded" sounds strange; revise it.

          Lines 380 and 381: I suggest changing "are considered" to "seem".

          Line 382: Change "toxicity" and "suggested" to "toxic" and "suggests", respectively.

          Lines 383 and 384: The authors need to provide additional information on "a single component vaccine" and relevant references.

          Line 385: I would recommend "strategy" instead of "target".

Materials and Methods

          Line 395: Change "1/2a serotype" to "serotype 1/2a".

          Line 396: Add a parenthesis after "Seoul, Korea".

          Line 396: Clarify "freshly grown".

          Lines 397-399: Were all the components of BHI powder mixed in the lab? If so, show the manufacturer information for each ingredient. If not, the composition of BHI is not necessary and show the manufacturer of BHI.

          Line 400: Change "measuring the absorbance spectrophotometrically" to "spectrophotometrically measuring the absorbance". Show the model no. and manufacturer of the spectrophotometer.

          Lines 403 and 404: Provide the reference for MIC determination protocol.

          Line 404: Change "medium adjusted" to "medium and adjusted".

          Lines 405 and 406: What does "these chemicals" mean? If "these chemicals" refers to HCl, H2SO4 and NaOH, I wonder why unit mg/mL was utilized since molar concentration is usually used for these chemicals. Show the volumes of the bacterial culture and chemicals that were employed in this assay.

          Lines 407-409: Why were LB agar plates used here? Was OD600 not employed in determining MIC?

          Lines 414-416: Revise "Stock solution ... MICs" for clarity. Why was the distilled water added last? Cells might have been exposed to excessive concentration of a chemical, albeit for a short period of time.

          Lines 421 and 442: Was temperature controlled during centrifugation? If so, specify the temperature.

          Line 422: Change "re-suspended" to "resuspended".

          Lines 425 and 426: Provide the manufacturer information for glutaraldehyde.

          Lines 438-440: Explain how this concentration DNA was analyzed.

          Lines 440-450: This part pertains to total protein analysis and I recommend starting a new paragraph from "In addition".

          Line 454: Revise "different chemicals treated LMGs" for clarity.

          Line 455: Explain why 16S rRNA and iap genes were selected for this analysis.

          Line 459: Change "initiated" to " were initiated".

          Line 468: Show the manufacturer of "Mxpro software".

          Line 470: Change "macrophage (KCLB:40071, RAW 264.7) cells" to "macrophage cells (KCLB:40071 and RAW 264.7)".

          Line 475: Change the comma next to "5 and 50 μg/mL" with a semi-colon.

          Line 477: Change the comma next to "CCK-8" with a semi-colon.

          Line 479: Change "% Cytotoxicity" to "% cytotoxicity".

          Line 486: Move "(Takara Bio, Shiga, Japan)" next to "RNAiso Plus" in line 485.

          Line 495: What is "Tli RNaseH Plus"? "Shiga, Japan" is not necessary here since it was already shown.

          Line 497: Show the manufacturer of "Stratagene Mx3005P cycler".

          Lines 506-514: Show the volume of PBS, LMGs, and bacterial culture injected into mice.

          Line 529: What is the model no. of the microplate reader?

          Line 532: Change "were loaded" to "was loaded".

          Line 548: Change "by fluorescence-activated cell sorter (FACS) analysis" to "with fluorescence-activated cell sorter (FACS) analysis".

          Lines 555 and 556: Double-check the equation since "100" instead of "1" should be used here.

          Line 559: Provide the model no. and manufacturer of "tissue homogenizer".

          Lines 563 and 564: Change "statistical significance one-way ANOVA" to "statistical significance with one-way ANOVA". Show the manufacturer of SPSS.

          Line 564: Change "are expressed" to "were expressed".

Conclusions

          Line 570: Change "by MICs" to "at MICs".

          Lines 571 and 572: Provide references for "other known methods".

          Line 572: Change "to produce" to "to producing".

Figures and Tables

Figure 2

          Lines 146 and 147: Change "chemically induced LMGs by" to "LMGs chemically induced by".

          Line 147: "M, 1 kb Marker Ladder" could be mentioned separately since it appears in all the agarose gel pictures.

          Line 148: Revise "respective chemical treated".

          Lines 149 and 150: Elaborate "Remaining protein profiles".

          Line 150: Add "and" in front of "HCl-".

          Line 152: Mention that Figure 2f shows proteins present in culture supernatants.

Figure 3

          Line 180: I suggest changing "remaining" to "remnant".

          Lines 180 and 181: Modify "Amplified (a) 16S rRNA and (b) iap gene by using" to "(a) 16S rRNA and (b) iap genes amplified with".

          Line 182: What does "TE-buffer; E. coli; and S. enteriditis (negative controls)" mean? Re-write it. 

Figure 4

          Line 196: Change the commas with semi-colons.

Figure 5

          Line 258: The authors used E. coli and Escherichia coli in figure legends. Keep the term consistent.

          Line 261: Does "triplicate experiments" mean "three independent experiments"?

          Lines 262 and 263: "Mean separation ... P<0.05." is fragmentary. Revise it.

Figure 6

          Lines 293 and 294: Change "PBS treated control" to "PBS-treated control".

          Line 299: Add "(P < 0.05)" after "non-immunized groups".

Figure 7

          Line 349: Elaborate "corresponding analysis".

          In the figure, CD4+- and CD8+-T cells are not clearly shown. Also, what do red and green colors represent in the first three rows? What do squares and the letters inside them represent?

          I suggest designating the first three rows and the last row to Figures 7a and 7b, respectively.

Figure 8

          It would be more informative if the authors divide the figure by rows, not by columns.

          The legend mostly concerns the first row, which shows the viable cell counts, and does not give any information on the data shown in the last two rows.

Figure 9

          Line 389: Change "were immunized" to "immunized".

          Lines 390 and 391: Change "performed" to "were performed".

          Lines 391 and 392: Re-write the statement "Overall ... test" for clarity.

Table 2

          Include target genes and references.

Author Response

Manuscript ID: ijms-1447112

Title: Protective immunity against Listeria monocytogenes in rats, provided by HCl- and NaOH-induced Listeria bacterial ghosts (LMGs) as vaccine candidates

Reviewer 2

Comments and Suggestions for Authors

Ji et al. generated Listeria monocytogenes bacterial ghosts (LMGs) using different acidic and alkaline chemicals (HCl, H2SO4 and NaOH) and demonstrated that these LMGs could elicit immune responses and protect mice against L. monocytogenes infection. This study is well designed and the potential of LMGs as vaccine was thoroughly investigated using various methods. However, the current manuscript would benefit from further revisions that are suggested below.

1. Although chemically generated LMGs are heavily examined in this study, it is not clear why this investigation is necessary, i.e., what novelty this study can bring. My understanding is that NaOH was already employed to generate BGs from L. monocytogenes, which successfully stimulated immune response and protected against L. monocytogenes. Then, why is generating BGs with acids important if the alkaline procedure is already working fine?

Response: We really thank the reviwer’s valuable comments. There is two methods to produce bacterial ghosts. One is lysis E-gene induced BGs and another one is chemically induced BGs. Lysis E gene induced BGs failed to produce BGs from Gram-positive bacteria due to thick cell wall. It has been proven that NaOH can able to make BGs from Gram-positive bactria and so far only chemical can able to make Gram-postive BGs. Therefore still need further improvement and we hypothesis HCl induced BGs from Gram-positive bacteria may have superior activities than NaOH induced BGs due to acid properities. As we expected, HCl induced LMGs showed better results than NaOH induced LMGs in specific in vivo experiments. Especially, bacterial load in liver (Figure 8) and survival analysis (Figure 9).

2. Discussion is rather weak and should be carefully revised. The most obvious weakness is that, while many prior studies were described in Results and Discussion, it was often not clear why they were described and how they were related to the data. Specific examples are as follows:

Lines 233-237 and 240-242: I cannot understand what purpose these statements serve here. What information do the authors want to convey via these statements?

Response: Thank you for indicating us. We have removed those statements and revised the  manuscript as per reviewer’s suggestion.

Lines 283-290 & 338-346: It is not clear how these statements are related to the data obtained in this study.

Response: Thank you for indicating us. We have removed those statements and revised the  manuscript as per reviewer’s suggestion.

3. Lines 215-229: This part should be further streamlined to convey what these results indicate and why they are important.

Response: Thank you for indicating us. We have revised extensively to simplify the results. Besides, we have analyzed cytokine mRNA expression using P values and changed to a new Figure 5 and legend.

4. Although LMGs were created with three different chemicals, i.e., HCl, H2SO4 and NaOH, all the immunological and animal experiments were conducted with HCl- and NaOH-induced LMGs. Why were these selected for those experiments?

Response: Thank you for the comment. Based on our in vitro results, NaOH and HCl induced LMG showed better results than H2So4 induced LMG (Figure 1 & 2). That’s why we have chosen HCl induced LMGs for further analysis.

5. Why was strain KCCM 40307 used in this study instead of the extensively studied lab strains? Which CC/ST does this strain belong to? Is the whole genome sequence available for this strain?

Response: KCCM 40307 is also pathogenic bacteria and have been extensively studied by many other researchers. We have mentioned few more references for reviewer’s kind perusal:

  • Park J, Oh JH, Kang HK, Choi MC, Seo CH, Park Y. Scorpion-venom-derived antimicrobial peptide Css54 exerts potent antimicrobial activity by disrupting bacterial membrane of zoonotic bacteria. Antibiotics. 2020, 9(11):831
  • Lee H, Yoon H, Ji Y, Kim H, Park H, Lee J, et al. Functional properties of Lactobacillus strains isolated from kimchi. Int. J. Food Microbiol. 2011, 145, 155–161.
  • Sohn H, Chang Y.H, Yune J.H, Jeong C.H, Shin D.M, Kwon H.C, Kim D.H, Hong S.W, Hwang H, Jeong J.Y. Probiotic Properties of Lactiplantibacillus plantarum LB5 Isolated from Kimchi Based on Nitrate Reducing Capability. Foods 2020, 9, 1777.

6. More details should be provided for experimental procedures. For instance, manufacturer information lacks for multiple chemicals and equipment, and volumes of substances that were administered to mice are not shown as well. Also, in line 553, what exactly does "selective media" mean?

Response: Thank you for indicating us. We have added manufacturer information, volumes that were administered to mice and selective media in the revised manuscript as per reviewer’s suggestion.

7. Unclear statements abound and need to be edited to improve clarity. See specific comments.

Response: Thank you for indicating us. We have removed those statements and revised the  manuscript as per reviewer’s suggestion.

8. Cite figures and tables more frequently. See specific comments for some examples.

Response: Thank you for indicating us. We have removed the statement and revised the  manuscript as per reviewer’s suggestion.

9. Provide references whenever necessary. Take a look at specific comments for details.

Response: Thank you for indicating us. We have removed the statement and revised the  manuscript as per reviewer’s suggestion.

10. Figures and their legends need to be re-worked. See specific comments.

Response: Thank you for indicating us. We have removed the statement and revised the  manuscript as per reviewer’s suggestion.

11. Provide strain names for E. coli, and S. enteritidis. Show the full name for "S. enteritidis". If this pertains to Salmonella, Change "S. enteritidis" to "S. Enteritidis" with "Enteritidis" unitalicized since "Enteritidis" is a serovar. Also, "S. enteriditis" seems to be a typo for "S. enteritidis". 

Response: We added strain (DH5α) of E. coli and revised S. enteritidis to S. enterica Enteritidis in the text.

Specific comments:

Abstract

1. Lines 17 and 18: Revise the statement since "observed" seems to be redundant with "proved".

Response: Thank you for this comment. We have revised the manuscript as per your suggestion.

2. Lines 21-23: Show what this statement indicates.

Response: It shows cytokine expression in the macrophages exposed to NaOH-induced LMGs and HCl-induced LMGs.

3. Line 26: Change "bactericidal activities" to "bactericidal activities of serum".

Response: It has been changed as “bactericidal activities of serum”

Introduction

4. Line 38: Change "gram-positive" to "Gram-positive".

Response: It has been changed to “Gram-positive”

5. Line 39: Change "pathogens of food-borne infection" to "foodborne pathogens".

Response: It has been changed to “foodborne pathogens”

6. Line 40: Change "at the cell envelope" to "in the cell envelope".

Response: It has been changed to “in the cell envelope”

7. Lines 43-46: Revise this sentence to improve clarity.

Response: We have changed this in the revised manuscript.

8. Line 48: Change "a host cell receptor known as" to "the host cell receptor".

Response: It has been changed to “the host cell receptor”

9. Lines 48-50: Why are these four proteins selected among many virulence factors involved in L. monocytogenes infection? Any particular reason? Also, I recommend changing "such as" to, "i.e.," since "such as" indicates that the listed proteins are a portion of four virulence factors, which is not the case here. Provide references.

Response: Thank you for the suggestion. Those four proteins are extensively studied and involved in the L. monocytogenes infections. We have wrote after the four proteins namnes in detail in the introduction section. Also, we have provided references on specific virulence genes in the revised manuscript as per reviewer’s suggestion.

10. Line 54: InlB has GW repeats but does not harbor LPXTG motif. Take a look at 2009 paper by Bonazzi et al. titled "Listeria monocytogenes Internalin and E-cadherin: From Bench to Bedside."

Response: Thank you for your valuable comment. I made a mistake. We deleted ‘Another LPTGX protein’ in the sentence and changed it to ‘InlB retaining three C-terminal glycine and tryptophan (GW) repeat domains is also required for Lm entry into epithelial cells.’

11. Lines 60 and 61: Provide additional references for hypervirulent strains since references 18 and 19 only pertain to antibiotic resistance.

We added a reference.

Response: (Raschle, S.; Stephan, R.; Stevens, M.J.A.; Cernela, N.; Zurfluh, K.; Muchaamba, F.; NÈ•esch-Inderbinen, M. Environmental dissemination of pathogenic Listeria monocytogenes in flowing surface waters in Switzerland. Sci. Rep. 2021, 11, 9066.)

12. Lines 61-63: Provide references.

[21] reference : A cross-protective vaccine against 4b and 1/2b Listeria monocytogenes.

13. Lines 63 and 230: Change "triple-virulence-genes deletion mutant" to "triple-virulence-gene deletion mutant".

Response: It has been changed to “triple-virulence-gene deletion mutant”

14. Line 66: Change "deliver" and "bind" to past tense.

Response: We have changed this in the revised manuscript.

15. Line 67: Change "platform as" to "platform for".

Response: It has been changed to “platform for”

16. Line 74: Change "forming" to "possessing".

Response: It has been changed to “possessing”

17. Lines 75 and 76: Change "retain" and "lacking" to "retained" and "lacked", respectively.

Response: It has been changed to “retained” and “lacked”

18. Line 81: Clarify "100% lysis rate". Does this mean the state of being completely lysed?

Response: yes, 100% lysis means no viable cells after BGs production.

19. Line 82: Provide further information for "potential risks".

Response: For bacterial infection, few number of viable cells are needed to couse mild to severe infections therefore 100% lysis rate is important to avoid risks.

20. Lines 83 and 84: Revise "sodium ... (MIC)" for clarity. Also, the full name for MIC is "minimum inhibitory concentration".

Response: We have changed as you suggested.

21. Lines 88 and 89: Change "strongly bactericidal effects" to "strong bactericidal effects". Change "are applied" to past tense and provide references.

Response: We have changed the statement and added reference in the revised manuscript.

22. Lines 89-91: This statement needs to be reworked for clarity.

Response: We have revised in the manuscript.

23. Lines 91 and 92: Revise "whether BGs ... alkali treatment" for clarity.

Response: We have changed this in the revised manuscript.

24. Line 98: "some" sounds vague; revise it.

 Response: We have changed this in the revised manuscript.

Results and Discussion

  1. Line 120: Double-check whether Figure 2e is suitable here since figure 2e shows proteins.

Response: Thank you for this comment. It was typo error (it’s Figure 2c). We have changed in text.

  1. Lines 121-123: Cite relevant figures.

Response: We have inserted relevant figure number in the revised manuscript. Figure 2d.

  1. Line 128: Change "HCl-" to "HCl".

Response: We have changed this in the revised manuscript.

  1. Line 129: Why do the authors think that the protein profile is due to "absence of cytoplasmic proteins"? Provide a further explanation and, if available, relevant references.

Response: Numerous stidies have suggested that BGs are intact cytoplasm-free. Besides, transmission electron microscopy analyses have revealed that BGs show a loss of cytoplasmic contents (empty bacterial envelopes) and maintain the cellular morphology similar to native bacteria where all cell surface structures including outer membrane proteins, adhesins, LPS and peptidoglycan layer are preserved. We have added some sentences and references supporting “absence of cytoplasmic proteins

References

1)  Ma, Y.; Cui, L.; Wang, M.; Sun, Q.; Liu, K.; Wang, J. A novel and efficient high-yield method for preparing bacterial ghosts. Toxins 2021, 13, 420.

2)  Lim, J.; Koh, V.H.Q.; Cho, S.S.L.; Periaswamy, B.; Choi, D.P.S.; Vacca, M.; De Sessions, P.F.; Kudela, P.; Lubitz, W.; Pastorin, G.; Alonso, S. Harnessing the immunomodulatory properties of bacterial ghosts to boost the anti-mycobacterial protective immunity. Front. Immunol. 2019, 10, 2737.

3)  Hajam, I.A.; Dar, P.A.; Appavoo, E.; Kishore, S.; Bhanuprakash, V., Ganesh, K. Bacterial ghosts of Eschericia coli drive efficient maturation of bovine monocyte-derived dendritic cells. PLos ONE 2015, 10, e0144397.

4)  Langemann, T.; Koller, V.J.; Muhammad, A.; Kudela, P.; Mayr, U.B.; Lubitz, W. The bacterial ghost platform system: Production and applications. Bioeng. Bugs. 2010, 1, 326–336.

  1. Lines 130-132: Change "extracted from" to "between". Cite relevant figures.

Response: We have changed this in the revised manuscript.

  1. Line 132: Change "suggested" to "suggests".

Response: We have changed this in the revised manuscript.

  1. Line 135: Change "showed" to "that showed".

Response: We have included “that showed” this in the revised manuscript.

  1. Line 160: Change "are 34.07" to "were 34.07". Revise other similar instances accordingly.

Response: We have changed this in the revised manuscript.

  1. Line 161: Elaborate "no Ct".

Response: “no Ct” those samples showed no cyle of of threshold and were therefore not detectd by Real-Time qPCR.

  1. Line 171: Change "based on" to "at".

Response: We have changed this in the revised manuscript.

  1. Lines 172 and 173: This statement could be further abbreviated. 

Response: We have revised this in the text.

  1. Lines 173-175: This statement is rather redundant and should be placed in the prior paragraph. Cite figures instead of "As shown in Figures 3a and 3b".
  2. Line 184: Change "The cytotoxicity" to "Cytotoxicity".

Response: We have changed this in the revised manuscript.

  1. Line 186: Please show why E. coli LPS was used in this assay.

Response: We used commercial LPS from E. coli O111:B4 (Millipore Sigma) as a control in cytoxicity assay and RT-qPCR.

  1. Line 189: Change "It suggested" to "It suggests".

Response: We have changed this in the revised manuscript.

  1. Line 190: Show the concentrations tested here.

Response: We have inserted tested concentrations in the revised manuscript.

  1. Lines 191-192, 210, and 317: Change "In this respect"; this transition word does not sound natural.

Response: We have changed this in the revised manuscript.

  1. Lines 203 and 204: Provide references for "Some reports". I recommend changing "Some" to "Several".

Response: We have changed and provided references in the revised manuscript.

  1. Lines 215-218: Cite relevant figures and remove "In this study".

Response: We have added figure number and removed “In this study” in the revised manuscript.

  1. Lines 221 and 223: Remove "Among treatments".

Response: We have changed this in the revised manuscript.

  1. Lines 230 and 231: Clarify whether "vaccine strain" is L. monocytogenes or not. If so, which strain?

Response: Thank you for this comment. Reference 21 (Meng et al, 2020): according to reference 21, highly attenuated Lm NTSNΔactA/plcB/orfX is called vaccine strain. They have used NTSN (4b serotype Lm) to make attenuated trible mutant Lm vaccine strain candidate.

  1. Line 233: Change "group. [20]." to "group [20]."

Response: We have changed this in the revised manuscript.

  1. Lines 238 and 239: Cite figures and change "post-exposure" to "post exposure".

Response: We have added figure number and changed to “post exposure” in the revised manuscript.

  1. Lines 245-247: Provide references.

Response: We added a reference.

(Castro, F.; Cardoso, A.P.; Gonçalves, R.M.; Serre, K.; Oliveira, M.J. Interferon-Gamma at the Crossroads of Tumor Immune Surveillance or Evasion. Front. Immunol. 2018, 9, 847.)

  1. Line 252: Change "suggested" to "suggests".

Response: We have changed this in the revised manuscript.

  1. Line 255: Change "to induce" to "in inducing".

 Response: We have changed this in the revised manuscript.

  1. Lines 265 and 266: I recommend citing Figure 6a rather than Figure 6.

Response: We have changed to figure 6a in the revised manuscript.

  1. Line 266: Change "On week 2" to "At week 2" and revise other similar instances as well.

 Response: We have changed this in the revised manuscript.

  1. Line 269: Clarify "significantly high". Higher than what?

 Response: Both HCL-induced LMGs and NaOH-induced LMGs were significantly higher than the PBS-injected control at week 8. We have we-rote the sentence in the revised manuscript.

  1. Line 270: Change "3.7 fold" to "3.7-fold" and revise similar instances as well.

 Response: We have changed this in the revised manuscript.

  1. Lines 273 and 274: Re-write the statement "Antisera ... virulent Lm" to improve clarity.

 Response: We have re-wrote the statement in the revised manuscript.

  1. Lines 283-285: Provide references.

Response: We added a reference.

(Zenewicz, L.A.; Shen, H. Innate and adaptive immune responses to Listeria monocytogenes: A short overview. Microbes Infect. 2007, 9, 1208–1215.)

  1. Line 303: Change "a great necessity" to "of great necessity".

 Response: We have changed this in the revised manuscript.

  1. Lines 304 and 305: Revise the statement for clarity.

 Response: We have removed this in the revised manuscript.

  1. Lines 305-307: Clarify "antisera ... respectively" and cite figures.

Response: We have changed this in the revised manuscript.

  1. Lines 308 and 309: Revise the statement for clarity. Also, it would be helpful if the authors show the size of six antigenic proteins commonly found in two LMGs.

Response: We have revised the statement for clarity and also described the size of six proteins.

  1. Line 311: Change "80 kDa protein" to "a 80 kDa protein" and revise a similar instance in line 312.

Response: We have corrected this in the revised manuscript.

  1. Line 313: Change "showed similar sizes as" to "were of similar sizes to". Provide references for InlA and InlB.

Response: We have changed this in the revised manuscript.

InlA (Franciosa, G.; Maugliani, A.; Scalfaro, C.; Floridi, F.; Aureli, P. Expression of internalin A and biofilm formation among Listeria monocytogenes clinical isolates. Int. J. Immunopathol. Pharmacol. 2009, 22, 183-193.)

InlB (Liu, D. Identification, subtyping and virulence determination of  Listeria monocytogenes, an important foodborne pathogen. J. Med. Microbiol. 2006, 55, 645-659.)

  1. Lines 313 and 314: Change "corresponding sizes of" to "of sizes corresponding to". Provide references for LAP, ActA, and LLO.

Response: We have changed this in the revised manuscript.

LAP (Jaradat, Z.W.; Wampler, J.W.L.; Bhunia, A.W.L.K. A Listeria adhesion protein-deficient Listeria monocytogenes strain shows reduced adhesion primarily to intestinal cell lines. Med. Microbiol. Immunol. 2003, 192, 85-91.)

ActA  (Niebuhr, K.; Chakraborty, T.; Rohde, M.; Gazlig, T.; Jansen, B.; Köllner, P.; Wehland, J. Localization of the ActA polypeptide of  Listeria monocytogenes in infected tissue culture cell lines: ActA is not associated with actin “comets”. Infect. Immun. 1993, 61, 2793-2802.)

LLO (Liu, D. Identification, subtyping and virulence determination of  Listeria monocytogenes, an important foodborne pathogen. J. Med. Microbiol. 2006, 55, 645-659.)

  1. Lines 315 and 316: Change "It is speculated" to "It was speculated". Revise "some relevant antigenic proteins" since it sounds vague.

Response: We have changed this in the revised manuscript.

  1. Line 317: Change "such as" to "such as those of".

Response: We have changed this in the revised manuscript.

  1. Lines 317-321: Revise it for clarity.

Response: We deleted the sentence including references for the clarity.

  1. Line 322: Change "Escherichia coli" to "E. coli". 

Response: We have changed this in the revised manuscript.

  1. Lines 323-325: I do not think that the detailed information on truncated InlB fragments such as "F3 fragment = cap + LRR" is necessary.

Response: We have changed this in the revised manuscript.

  1. Line 327: Provide a reason for FACS analysis instead of "In the present study".

Response: We have changed this in the revised manuscript.

  1. Lines 333 and 334: In which was the percent of SigM-B cell population higher? Be more specific in describing the data.

Response: We have re-wrote the statement in the revised manuscript.

  1. Lines 335 and 336: Change "suggested" to "suggests".

Response: We have changed this in the revised manuscript.

  1. Lines 339 and 340: Revise "phagocytosis-mediated ... mucosa" to improve clarity.

Response: We have changed this into phagocytosis-mediated opsoniztion in the revised manuscript.

  1. Line 341: Change "into a cytoplasm" to "into cytoplasm".

 Response: We have changed this in the revised manuscript.

  1. Line 353: Remove "In this study".

Response: We have changed this in the revised manuscript.

  1. Line 356: Revise "at two weeks post-challenge" to improve clarity.

 Response: We have changed this in the revised manuscript.

  1. Line 357: Clarify "vaccinated with ... LMGs".

 Response: We have changed this in the revised manuscript.

  1. Line 362: Change "Another finding" and "anti-ActA and anti-LLO antibodies administration" to "Another study" and "anti-ActA and anti-LLO antibody administration".

 Response: We have changed this in the revised manuscript.

  1. Lines 371-378: This part should be placed at the beginning of section 2.10, i.e., line 353.

 Response: We have changed this in the revised manuscript.

  1. Lines 372 and 373: Clarify "showed ... concentrations of Lm".

Response: We have used various concentration of Lm to determine the infection studies after vaccination.

  1. Line 373: Change "At 11 days post-inoculation" to "At day 11 post inoculation".

Response: We have changed this in the revised manuscript.

  1. Line 374: Provide specific concentrations tested here after "others".

Response: We have described specific concentrations in the revised manuscript.

  1. Lines 375 and 376: This statement is vague and will confuse readers. What data are presented in Figure 9b? "Significant differences" in what? If "significant differences" are present, which group showed a higher value?

Response: We revised the sentence for clarification.

  1. Lines 376-378: Move this statement to the next paragraph. Also, "concluded" sounds strange; revise it.

Response: Thank you for your suggestion. We have moved the sentence to the next paragraph in the revised manuscript.

  1. Lines 380 and 381: I suggest changing "are considered" to "seem".

Response: We have changed this in the revised manuscript.

  1. Line 382: Change "toxicity" and "suggested" to "toxic" and "suggests", respectively.

Response: We have changed this in the revised manuscript.

  1. Lines 383 and 384: The authors need to provide additional information on "a single component vaccine" and relevant references.

Response: We have added additional information and a relevant reference in the revised manuscript.

(Saul, A.; Fay, M.P. Human immunity and the design of multi-component, single target vaccines. PLoS ONE 2007, 2, e850.)

  1. Line 385: I would recommend "strategy" instead of "target".

 Response: We have changed this in the revised manuscript.

Materials and Methods

  1. Line 395: Change "1/2a serotype" to "serotype 1/2a".

Response: We have changed this in the revised manuscript.

  1. Line 396: Add a parenthesis after "Seoul, Korea".

Response: We have changed this in the revised manuscript.

  1. Line 396: Clarify "freshly grown".

Response: Yes, culture was not stored in refrigerator and freshly grown before bacterial ghosts preparation.

  1. Lines 397-399: Were all the components of BHI powder mixed in the lab? If so, show the manufacturer information for each ingredient. If not, the composition of BHI is not necessary and show the manufacturer of BHI.

Response: We have added manufacturer of BHI in the revised manuscript.

  1. Line 400: Change "measuring the absorbance spectrophotometrically" to "spectrophotometrically measuring the absorbance". Show the model no. and manufacturer of the spectrophotometer.

Response: We have changed this in the revised manuscript.

  1. Lines 403 and 404: Provide the reference for MIC determination protocol.

Response: We have aadded reference in the revised manuscript.

  1. Line 404: Change "medium adjusted" to "medium and adjusted".

Response: We have changed this in the revised manuscript.

  1. Lines 405 and 406: What does "these chemicals" mean? If "these chemicals" refers to HCl, H2SO4 and NaOH, I wonder why unit mg/mL was utilized since molar concentration is usually used for these chemicals. Show the volumes of the bacterial culture and chemicals that were employed in this assay.

Response: Yes, these chemicals refer to HCl, H2SO4 and NaOH and we have changed this in the revised manuscript. Regarding unit (mg/mL), we have followed previously published our research articles and other researcher articles for reference. We have mentioned some of the references for your kind perusal.

Our published articles:

  1. Vinod, N.; Oh, S.; Kim, S.; Choi, C.W.; Kim, S.C.; Jung, C.-H. Chemically induced Salmonella enteritidis ghosts as a novel vaccine candidate against virulent challenge in a rat model. Vaccine 2014, 32, 3249–3255.
  2. Vinod, N.; Oh, S.; Park, H.J.; Koo, J.M.; Choi, C.W.; Kim, S.C. Generation of a Novel Staphylococcus aureus Ghost Vaccine and Examination of Its Immunogenicity against Virulent Challenge in Rats. Immun. 2015, 83, 2957–2965.
  3. Park, H.J.; Oh, S.; Vinod, N.; Ji, S.; Noh, H.B.; Koo, J.M.; Lee, S.H.; Kim, S.C.; Lee, K.-S.; Choi, C.W. Characterization of chemically-induced bacterial ghosts (BGs) using sodium hydroxide-induced Vibrio parahaemolyticus ghosts (VPGs). J. Mol. Sci. 2016, 17, 1904.

Other research articles

  1. Lee, M.K., Park, B.K., Jeong, C.K. Oh, D.H. Antimicrobial activity of glycerol monolaurate and organic acids on the survival of Escherichia coli O157:H7. Food Sci. Nutr. 6: 6-9 (2001).
  2. Lallemand, E. A., Lacroix, M. Z., Toutain, P.-L., Boullier, S., Ferran, A. A., Bousquet-Melou, A. (2016). In vitro degradation of antimicrobials during use of broth microdilution method can increase the measured minimal inhibitory and minimal bactericidal concentrations. Frontiers in Microbiology7, 2051. 
  3. Lines 407-409: Why were LB agar plates used here? Was OD600 not employed in determining MIC?

Response: To double check, we have spread culture on to LB agar plates for verfification. Also, we have used OD600 for MIC determination.

  1. Lines 414-416: Revise "Stock solution ... MICs" for clarity. Why was the distilled water added last? Cells might have been exposed to excessive concentration of a chemical, albeit for a short period of time.

Response: To adjust specific concentration, we have used water and adjusted for various BG production. We have been following this method for various BG production. So far, we couldn’t faced any problem.

  1. Lines 421 and 442: Was temperature controlled during centrifugation? If so, specify the temperature.

Response: We have used room temperature centrifugation and revised in the text.

  1. Line 422: Change "re-suspended" to "resuspended".

Response: We have changed this in the revised manuscript.

  1. Lines 425 and 426: Provide the manufacturer information for glutaraldehyde.

Response: We have added the manufacturer in the revised manuscript.

  1. Lines 438-440: Explain how this concentration DNA was analyzed.

Response: In agarose gel electrophoresis, we didn’t measure the DNA concentration of LMGs, but visually determined the presence or absence of DNA band compared with the wild-type control.

  1. Lines 440-450: This part pertains to total protein analysis and I recommend starting a new paragraph from "In addition".

Response: We have changed this in the revised manuscript.

  1. Line 454: Revise "different chemicals treated LMGs" for clarity.

Response: We have changed this in the revised manuscript.

  1. Line 455: Explain why 16S rRNA and iap genes were selected for this analysis.

Response: We have used those genes based on previous studies and also we have added references.

  1. Line 459: Change "initiated" to " were initiated".

Response: We have changed this in the revised manuscript.

  1. Line 468: Show the manufacturer of "Mxpro software".

Response: We have added the manufacturer in the revised manuscript.

  1. Line 470: Change "macrophage (KCLB:40071, RAW 264.7) cells" to "macrophage cells (KCLB:40071 and RAW 264.7)".

Response: We have changed this in the revised manuscript.

  1. Line 475: Change the comma next to "5 and 50 μg/mL" with a semi-colon.

Response: We have changed this in the revised manuscript.

  1. Line 477: Change the comma next to "CCK-8" with a semi-colon.

Response: We have changed this in the revised manuscript.

  1. Line 479: Change "% Cytotoxicity" to "% cytotoxicity".

Response: We have changed this in the revised manuscript.

  1. Line 486: Move "(Takara Bio, Shiga, Japan)" next to "RNAiso Plus" in line 485.

Response: We have changed this in the revised manuscript.

  1. Line 495: What is "Tli RNaseH Plus"? "Shiga, Japan" is not necessary here since it was already shown.

Response: It is premix tag and product name (Tli RNaseH Plus). We have removed “Shiga, Japana” in the text.

  1. Line 497: Show the manufacturer of "Stratagene Mx3005P cycler".

Response: We have added the manufacturer in the revised text.

  1. Lines 506-514: Show the volume of PBS, LMGs, and bacterial culture injected into mice.

Response: We have changed this in the revised manuscript.

  1. Line 529: What is the model no. of the microplate reader?

Response: We have added the model no. and manufacturer in the revised manuscript.

  1. Line 532: Change "were loaded" to "was loaded".

Response: We have changed this in the revised manuscript.

  1. Line 548: Change "by fluorescence-activated cell sorter (FACS) analysis" to "with fluorescence-activated cell sorter (FACS) analysis".

Response: We have changed this in the revised manuscript.

  1. Lines 555 and 556: Double-check the equation since "100" instead of "1" should be used here.

Response: Thank you for your suggestion. We revised the equation correctly.

  1. Line 559: Provide the model no. and manufacturer of "tissue homogenizer".

Response: We have added the model no. and manufacturer in the revised manuscript.

  1. Lines 563 and 564: Change "statistical significance one-way ANOVA" to "statistical significance with one-way ANOVA". Show the manufacturer of SPSS.

Response: We have changed this in the revised manuscript.

  1. Line 564: Change "are expressed" to "were expressed".

 Response: We have changed this in the revised manuscript.

Conclusions

  1. Line 570: Change "by MICs" to "at MICs".

Response: We have changed this in the revised manuscript.

  1. Lines 571 and 572: Provide references for "other known methods".

Response: We have added references in the revised manuscript.

  1. Line 572: Change "to produce" to "to producing".

 Response: We have deleted this sentence according to another reviwers’ suggection.

Figures and Tables

Figure 2

  1. Lines 146 and 147: Change "chemically induced LMGs by" to "LMGs chemically induced by".

Response: We have changed this in the revised manuscript.

  1. Line 147: "M, 1 kb Marker Ladder" could be mentioned separately since it appears in all the agarose gel pictures.

Response: We have changed this in the revised manuscript.

  1. Line 148: Revise "respective chemical treated".

Response: We have revised this statement in the revised manuscript.

  1. Lines 149 and 150: Elaborate "Remaining protein profiles".

Response: We have elaborated this in the revised manuscript.

  1. Line 150: Add "and" in front of "HCl-".

Response: We have changed this in the revised manuscript.

  1. Line 152: Mention that Figure 2f shows proteins present in culture supernatants.

 Response: We have mentioned this in the revised manuscript.

Figure 3

  1. Line 180: I suggest changing "remaining" to "remnant".

Response: We have changed this in the revised manuscript.

  1. Lines 180 and 181: Modify "Amplified (a) 16S rRNA and (b) iap gene by using" to "(a) 16S rRNA and (b) iap genes amplified with".

Response: We have modified this in the revised manuscript.

  1. Line 182: What does "TE-buffer; E. coli; and S. enteriditis (negative controls)" mean? Re-write it. 

Response: Those are negative controls. We have re-wrote this in the revised manuscript.

Figure 4

  1. Line 196: Change the commas with semi-colons.

Response: We have changed this in the revised manuscript.

Figure 5

  1. Line 258: The authors used E. coli and Escherichia coli in figure legends. Keep the term consistent.

Response: We have changed this in the revised manuscript.

  1. Line 261: Does "triplicate experiments" mean "three independent experiments"?

Response: Yes, three independent experminets

  1. Lines 262 and 263: "Mean separation ... P<0.05." is fragmentary. Revise it.

Response: We have changed this in the revised manuscript.

Figure 6

  1. Lines 293 and 294: Change "PBS treated control" to "PBS-treated control".

Response: We have changed this in the revised manuscript.

  1. Line 299: Add "(P < 0.05)" after "non-immunized groups".

 Response: We have added this in the revised manuscript.

Figure 7

  1. Line 349: Elaborate "corresponding analysis".

Response: We have elaborated in the revised manuscript.

  1. In the figure, CD4+- and CD8+-T cells are not clearly shown. Also, what do red and green colors represent in the first three rows? What do squares and the letters inside them represent? I suggest designating the first three rows and the last row to Figures 7a and 7b, respectively.

Response: Red color areas represent reaction intensities between CD45R marker and SIgM (a), between CD3e marker and CD4 (b), and between CD4e marker and CD8. (b) P2 box indicates a strong reaction between CD3 marker and CD4. (c) P2 box and P3 box indicate CD4 reaction and CD 8 reaction, respectively. We revised Figure caption as you suggested.

Figure 8

  1. It would be more informative if the authors divide the figure by rows, not by columns. The legend mostly concerns the first row, which shows the viable cell counts, and does not give any information on the data shown in the last two rows.

 Response: We have revised figure and legends as you suggested.

Figure 9

  1. Line 389: Change "were immunized" to "immunized".

Response: We have changed this in the revised manuscript.

  1. Lines 390 and 391: Change "performed" to "were performed".

Response: We have changed this in the revised manuscript.

  1. Lines 391 and 392: Re-write the statement "Overall ... test" for clarity.

Response: We have re-wrote the staement in the revised manuscript.

 Table 2

  1. Include target genes and references.

Response: Primers and probes used in this study were designed based on sequences of each target gene from the NCBI database. We have revised Table 2.

Round 2

Reviewer 2 Report

          I appreciate the authors for addressing this reviewer's comments and incorporating many suggestions into the manuscript. However, some comments have been overlooked or insufficiently taken care of as shown below.

Major comments

          Comment 5: The response to comment 5 is rather not satisfactory since CC/ST information and the presence of whole genome sequence have not been mentioned in the rebuttal.

Comment 11: The strain name for Salmonella Enteritidis is not shown in the revised manuscript.

Introduction

          Comment 7: The revised sentence still needs to be amended to improve clarity.

          Comment 11: I do not think that the paper by Raschle et al. is not a good reference for hypervirulent strains and recommend the following 2016 paper by Maury et al.

          Maury MM, Tsai YH, Charlier C, Touchon M, Chenal-Francisque V, Leclercq A, Criscuolo A, Gaultier C, Roussel S, Brisabois A, Disson O, Rocha EPC, Brisse S, Lecuit M. Uncovering Listeria monocytogenes hypervirulence by harnessing its biodiversity. Nat Genet. 2016 Mar;48(3):308-313. doi: 10.1038/ng.3501. Epub 2016 Feb 1. Erratum in: Nat Genet. 2017 Mar 30;49(4):651. Erratum in: Nat Genet. 2017 May 26;49(6):970. PMID: 26829754; PMCID: PMC4768348.

          Comment 18: Incorporate the response to the text in the manuscript.

          Comment 19: I still think that some extra explanation for "potential risks" is needed for readers.

Results and Discussion

          Comment 29: In line 136 of the revised manuscript, change "extracted between" to "between".

          Comment 33: Incorporate the response to the text in the manuscript.

          Comment 34: In line 180 of the revised manuscript, change "based at" to "at".

          Comment 36: There is no response for this comment and "Furthermore" in line 182 of the revised manuscript sounds rather strange.

          Comments 43 and 49: No references were added although the authors claimed to have done so.

          Comment 54: I suggest incorporating the response to the text, which is not included in the revised manuscript.

          Comment 76: In lines 404 and 405, change "at 2 weeks post-challenge" to "at week 2 post challenge".

          Comment 77: In lines 405 and 406, change "homogenates of HCl- and NaOH-induced LMGs" to "homogenates of the mice vaccinated with HCl- and NaOH-induced LMGs".

          Comment 84: I cannot see any suggested changes although the authors claimed to have added to the revised manuscript.

Materials and Methods

          Comment 106: The suggested change is not found in line 504 of the revised manuscript.

          Comment 111: Change "various chemicals treated LMGs" to "various chemically treated LMGs" in line 517 of the revised manuscript.

          Comment 120: Remove the comma next to "Tli RNaseH Plus" with a semi-colon.

Figure 5

          Comment 145: Change "triplicate experiments" to "three independent experiments" in line 288 of the revised manuscript.

Figure 8

          Comment 151: This suggestion has been neglected although the authors claimed that they have addressed it.

Additional comments:

          In the meantime, the revised manuscript would benefit from addressing the following extra comments.

Line 22: Change "were observed" to "was observed".

Line 55: Re-write "retaining".

Line 92: Change "were amplified" to "was amplified".

Line 93: Change "arose regarding that" to "arose regarding whether".

Line 156: Change "While LMGs production" to "During LMGs production".

Line 207: Change the semi-colons separating concentrations to colons; and add "and" in front of "3.7*10^6".

Line 421: Change "significant differences in" to "significant differences at".

Line 458: Move "(Libra S22, Biochrom Ltd., Cambridge, UK)" to the end of the sentence.

Author Response

Round-2

I appreciate the authors for addressing this reviewer's comments and incorporating many suggestions into the manuscript. However, some comments have been overlooked or insufficiently taken care of as shown below.

Major comments

Comment 5: The response to comment 5 is rather not satisfactory since CC/ST information and the presence of whole genome sequence have not been mentioned in the rebuttal.

Response: The whole sequence of Lm KCCM 40307 has not been reported yet. Therefore, we have no idea of CC/ST information.

Comment 11: The strain name for Salmonella Enteritidis is not shown in the revised manuscript.

Response: A virulent strain of Salmonella Enteritidis was provided from an animal health product manufacturing company (KBNP, Inc., Korea). This is field strain, and we don’t have strain name.

Reference for Salmonella enteritidis strain which we used in this study:

Vinod, N.; Oh, S.; Kim, S.; Choi, C.W.; Kim, S.C.; Jung, C.-H. Chemically induced Salmonella enteritidis ghosts as a novel vaccine candidate against virulent challenge in a rat model. Vaccine 2014, 32, 3249–3255.

Introduction

Comment 7: The revised sentence still needs to be amended to improve clarity.

Response: We have changed this in the revised manuscript.

Comment 11: I do not think that the paper by Raschle et al. is not a good reference for hypervirulent strains and recommend the following 2016 paper by Maury et al.

          Maury MM, Tsai YH, Charlier C, Touchon M, Chenal-Francisque V, Leclercq A, Criscuolo A, Gaultier C, Roussel S, Brisabois A, Disson O, Rocha EPC, Brisse S, Lecuit M. Uncovering Listeria monocytogenes hypervirulence by harnessing its biodiversity. Nat Genet. 2016 Mar;48(3):308-313. doi: 10.1038/ng.3501. Epub 2016 Feb 1. Erratum in: Nat Genet. 2017 Mar 30;49(4):651. Erratum in: Nat Genet. 2017 May 26;49(6):970. PMID: 26829754; PMCID: PMC4768348.

Response: Thank you for the suggestion. We have replaced the reference in the revised manuscript as per your suggestion.

Comment 18: Incorporate the response to the text in the manuscript.

Response: We have changed this in the revised manuscript.

 Comment 19: I still think that some extra explanation for "potential risks" is needed for readers.

Response: For bacterial infection, few number of viable cells are needed to cause mild to severe infections. Incomplete lysis of bacteria may have potential risks associated with unwanted revert to virulence. Therefore, 100% complete lysis rate is important to avoid potential risks.

 Results and Discussion

Comment 29: In line 136 of the revised manuscript, change "extracted between" to "between".

Response: We have changed this in the revised manuscript.

Comment 33: Incorporate the response to the text in the manuscript.

Response: We have incorporated the response in the revised manuscript.

Comment 34: In line 180 of the revised manuscript, change "based at" to "at".

Response: We have changed this in the revised manuscript.

Comment 36: There is no response for this comment and "Furthermore" in line 182 of the revised manuscript sounds rather strange.

Response: Thank you for indicating us. We have removed those lines in the revised manuscript. Those statements already mentioned in the prior paragraph.

 Comments 43 and 49: No references were added although the authors claimed to have done so.

Response: Thank you for your suggestions.  We have included references.

Comment 54: I suggest incorporating the response to the text, which is not included in the revised manuscript.

Response: We have changed this in the revised manuscript.

Comment 76: In lines 404 and 405, change "at 2 weeks post-challenge" to "at week 2 post challenge".

Response: We have changed this in the revised manuscript.

Comment 77: In lines 405 and 406, change "homogenates of HCl- and NaOH-induced LMGs" to "homogenates of the mice vaccinated with HCl- and NaOH-induced LMGs".

Response: We have changed this in the revised manuscript.

Comment 84: I cannot see any suggested changes although the authors claimed to have added to the revised manuscript.

Response: We have removed that statement in the revised manuscript. The statement which we wrote will not fit in the next paragraph.

 Materials and Methods

Comment 106: The suggested change is not found in line 504 of the revised manuscript.

Response: We have changed this in the revised manuscript.

Comment 111: Change "various chemicals treated LMGs" to "various chemically treated LMGs" in line 517 of the revised manuscript.

Response: We have changed this in the revised manuscript.

Comment 120: Remove the comma next to "Tli RNaseH Plus" with a semi-colon.

Response: We have changed this in the revised manuscript.

Figure 5

Comment 145: Change "triplicate experiments" to "three independent experiments" in line 288 of the revised manuscript.

Response: We have changed this in the revised manuscript.

Figure 8

Comment 151: This suggestion has been neglected although the authors claimed that they have addressed it.

Response: Thank you for indicating us. We have changed figures as you suggested and revised figure legend also.

Additional comments:

In the meantime, the revised manuscript would benefit from addressing the following extra comments.

Line 22: Change "were observed" to "was observed".

Response: We have changed this in the revised manuscript.

Line 55: Re-write "retaining".

Response: We have changed this in the revised manuscript.

Line 92: Change "were amplified" to "was amplified".

Response: We have changed this in the revised manuscript.

Line 93: Change "arose regarding that" to "arose regarding whether".

Response: We have changed this in the revised manuscript.

Line 156: Change "While LMGs production" to "During LMGs production".

Response: We have changed this in the revised manuscript.

Line 207: Change the semi-colons separating concentrations to colons; and add "and" in front of "3.7*10^6".

Response: We believe that comma would be better, and I used comma. I don’t understand where to add “and” there is 3.7x10^6 two times in the text.

Line 421: Change "significant differences in" to "significant differences at".

Response: We have changed this in the revised manuscript.

Line 458: Move "(Libra S22, Biochrom Ltd., Cambridge, UK)" to the end of the sentence.

Response: We have changed this in the revised manuscript.